# SPATIALLADDER: PROGRESSIVE TRAINING FOR SPATIAL REASONING IN VISION-LANGUAGE MODELS

**Hongxing Li**[1,*], **Dingming Li**[1,*] **Zixuan Wang**[1] **Yuchen Yan**[1] **Hang Wu**[1]
**Wenqi Zhang**[1] **Yongliang Shen**[1,†], **Weiming Lu**[1], **Jun Xiao**[1] **Yueting Zhuang**[1]
[1]Zhejiang University
{hongxing.li, syl}@zju.edu.cn

 GitHub: https://github.com/ZJU-REAL/SpatialLadder
 Project: http://zju-real.github.io/SpatialLadder

## ABSTRACT

Spatial reasoning remains a fundamental challenge for Vision-Language Models (VLMs), with current approaches struggling to achieve robust performance despite recent advances. We identify that this limitation stems from a critical gap: existing methods attempt to learn spatial reasoning directly without establishing the hierarchical foundations of perception and understanding. To address this challenge, we present a comprehensive methodology for building spatial intelligence progressively. We introduce SpatialLadder-26$k$, a multimodal dataset containing 26,610 samples spanning object localization, single-image, multi-view, and video spatial reasoning tasks, constructed through a standardized pipeline that ensures systematic coverage across modalities. Building on this dataset, we design a three-stage progressive training framework that (1) establishes spatial perception through object localization, (2) develops spatial understanding through multi-dimensional spatial tasks, and (3) strengthens complex reasoning via reinforcement learning with verifiable rewards. This approach yields SpatialLadder, a 3B-parameter model that achieves state-of-the-art performance on spatial reasoning benchmarks, with 23.4% average improvement over the base model, surpassing GPT-4o by 20.8% and Gemini-2.0-Flash by 10.1%. Notably, SpatialLadder maintains strong generalization with 7.2% improvement on out-of-domain benchmarks, demonstrating that progressive training from perception to reasoning is essential for robust spatial intelligence.

## 1 INTRODUCTION

VLMs have achieved remarkable success in fundamental visual tasks (Huang et al., 2025; Yu et al., 2025), yet a critical capability remains elusive: spatial reasoning. While humans effortlessly grasp spatial relationships in visual scenes, current VLMs struggle even basic spatial queries (Yang et al., 2025a; Tong et al., 2024; Wu et al., 2025b). This limitation severely constrains their deployment in applications requiring spatial intelligence, from robotics navigation (Zitkovich et al., 2023) to autonomous driving (Tian et al., 2024) and virtual reality systems (Chandrasegaran et al., 2024).

The root cause of this spatial reasoning deficit lies in a fundamental gap between perception and reasoning in current VLM architectures (Chen et al., 2025; Li et al., 2025c). We hypothesize that existing approaches fail because they treat spatial reasoning as a monolithic capability, attempting to learn it directly from question-answer pairs without establishing the necessary hierarchical structure (Ouyang et al., 2025; Wu et al., 2025a). To validate this hypothesis, we conducted controlled experiments with 200 spatial orientation tasks, progressively adding perceptual hints to isolate the bottleneck (detailed in Appendix A). As shown in Figure 1, providing location hints (bounding boxes) improves accuracy by 5.0%, and additional directional cues yield another 4.5% gain, achieving 9.5% total improvement. This demonstrates that models possess latent reasoning capabilities but lack the perceptual grounding to activate them effectively. The primary bottleneck lies not in reasoning capacity but in the integration between perception and reasoning.

Current approaches to enhancing spatial reasoning in VLMs suffer from two fundamental limitations. First, existing datasets are fragmented and narrow in scope, focusing on either 2D images or

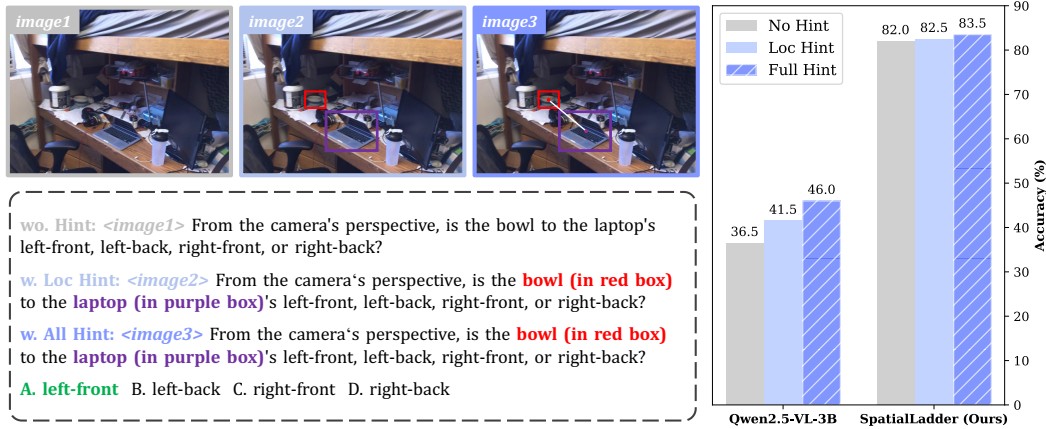

Figure 1: **Perception-reasoning gap in spatial reasoning.** Left: Three experimental conditions with increasing perceptual hints: no hints, location hints (bounding boxes), and full hints (boxes plus directional cues). Right: While Qwen2.5-VL-3B shows progressive improvement with increasing hints, our trained model achieves superior performance with negligible reliance on external prompts.

3D scenes in isolation (Liao et al., 2025; Ouyang et al., 2025; Kamath et al., 2023), while lacking systematic coverage across modalities and standardized annotation pipelines, resulting in incomplete training signals for comprehensive spatial understanding. Second, recent methods attempt to directly optimize reasoning outputs through reinforcement learning (Liao et al., 2025; Ouyang et al., 2025; Wu et al., 2025c) or auxiliary 3D representations (Wu et al., 2025a; Hong et al., 2023; Zhu et al., 2024), without establishing the hierarchical structure required for spatial intelligence: they bypass the critical progression from perceiving objects to understanding spatial relationships to performing logical inference, producing models that memorize patterns rather than develop genuine spatial understanding, leading to poor generalization on novel spatial configurations.

We address these challenges through a systematic approach based on the hierarchical nature of spatial intelligence. Our key insight is that robust spatial reasoning must be built progressively: establishing perceptual foundations through object localization, developing spatial understanding through multi-dimensional spatial analysis, and ultimately achieving complex reasoning through their integration.

To implement this vision, we introduce SpatialLadder-26$k$, a comprehensive multimodal dataset containing 26,610 samples across four complementary task categories: object localization (5,929 samples), single-image spatial reasoning (5,929 samples), multi-view spatial reasoning (5,752 samples), and video spatial reasoning (9,000 samples). Unlike existing datasets, SpatialLadder-26$k$ systematically covers the full spectrum from basic perception to complex reasoning. We develop a standardized pipeline leveraging 3D scene reconstructions from ScanNet to ensure consistent, high-quality annotations across all modalities.

Building on this dataset, we design a three-stage progressive training framework. Stage 1 establishes spatial perception through object localization tasks, teaching models to accurately identify and locate objects in scenes. Stage 2 develops spatial understanding through multi-dimensional tasks including size estimation, distance judgment, and orientation analysis across seven distinct spatial dimensions. Stage 3 employs Group Relative Policy Optimization (GRPO) (Shao et al., 2024) with task-specific verifiable reward functions to strengthen complex reasoning capabilities, enabling models to form coherent chains of spatial thought.

Through this progressive approach, we develop SpatialLadder, a 3B-parameter model that establishes new benchmarks in spatial reasoning performance. Extensive experiments demonstrate significant improvements: on VSI-Bench (Yang et al., 2025a), SpatialLadder achieves 45.7% accuracy. On our proposed SPBench-SI and SPBench-MV benchmarks, it attains 70.2% and 70.9% accuracy respectively. Across all benchmarks, SpatialLadder achieves an overall performance of 62.3%, surpassing the base model by 23.4% and outperforming GPT-4o by 20.8% and Gemini-2.0-Flash by 10.1%. Crucially, SpatialLadder maintains strong generalization with 7.2% average

improvement on out-of-domain benchmarks including CV-Bench (Tong et al., 2024), SPAR (Zhang et al., 2025), ViewSpatial-Bench (Li et al., 2025a), MMSI-Bench (Yang et al., 2025b) and MindCube (Yin et al., 2025), demonstrating the robustness of our progressive training approach.

Our contributions are threefold:

- We introduce SpatialLadder-26$k$, a comprehensive multimodal dataset with 26,610 samples spanning object localization and spatial reasoning across single-image, multi-view, and video modalities, constructed through a standardized pipeline ensuring systematic coverage and high-quality annotations.

- We design a three-stage progressive training framework that systematically builds spatial reasoning capabilities by establishing perceptual foundations, developing spatial understanding, and strengthening complex reasoning through reinforcement learning with verifiable rewards.

- We demonstrate that our approach yields significant performance improvements, with SpatialLadder achieving state-of-the-art results on multiple benchmarks while maintaining strong generalization to out-of-domain tasks, validating the effectiveness of progressive spatial learning.

## 2 RELATED WORKS

### 2.1 VISUAL SPATIAL REASONING

As a key capability of VLMs, visual spatial reasoning is more complex than general visual tasks and remains challenging (Yang et al., 2025a; Wu et al., 2025b). Despite notable advances in basic visual tasks (Li et al., 2024a;b; Tang et al., 2025a;b), extensive benchmark (Yang et al., 2025a; Wu et al., 2025b) evaluations demonstrate that they still face serious bottlenecks in spatial reasoning. Recent studies have attempted to explore multiple remedies, such as R1-Zero-VSI (Liao et al., 2025) and SpaceR (Ouyang et al., 2025), which utilize reinforcement learning to enhance models' spatial reasoning capabilities; Spatial-MLLM (Wu et al., 2025a), which introduces 3D representations (Wang et al., 2025a) as bridging knowledge; and Coarse Correspondences (Liu et al., 2025a), which improves models' spatiotemporal modeling capabilities through cross-frame object tracking. However, there remains a general lack of comprehensive, diverse, high-quality datasets, as well as effective training frameworks that advance from basic to complex concepts, to systematically enhance the capabilities of VLMs in spatial reasoning tasks.

### 2.2 REINFORCEMENT LEARNING IN VLMS

Recent studies have extended Reinforcement Learning (RL) techniques from LLMs to VLMs, leading to notable progress in visual reasoning (Liu et al., 2025b; Shen et al., 2025). Representative works such as Vision-R1 (Huang et al., 2025), MM-Eureka (Meng et al., 2025), and R1-OneVision (Yang et al., 2025c) have demonstrated that transferring RL methods to VLMs can significantly enhance the visual mathematical reasoning capabilities. In the video domain, Video-R1 (Feng et al., 2025) and VideoChat-R1 (Li et al., 2025b) applied RL to improve temporal understanding and video localization performance. Beyond text-oriented reasoning, methods like GRIT (Fan et al., 2025) and Pixel-Reasoner (Su et al., 2025a) leveraged RL to stimulate "thinking with images" (Su et al., 2025b), enabling models to perform structured and interpretable multimodal reasoning. Despite these advancements, research specifically targeting visual spatial reasoning remains limited. To address this gap, we propose a training paradigm that systematically enhances VLMs' capabilities in spatial reasoning tasks.

## 3 METHODS

We present a comprehensive framework that systematically builds spatial reasoning in VLMs via progressive training. Our approach consists of two core components: (1) SpatialLadder-26$k$, a multimodal dataset systematically spanning spatial tasks from basic perception to complex reasoning, and (2) a three-stage training framework reflecting the hierarchical nature of spatial intelligence.

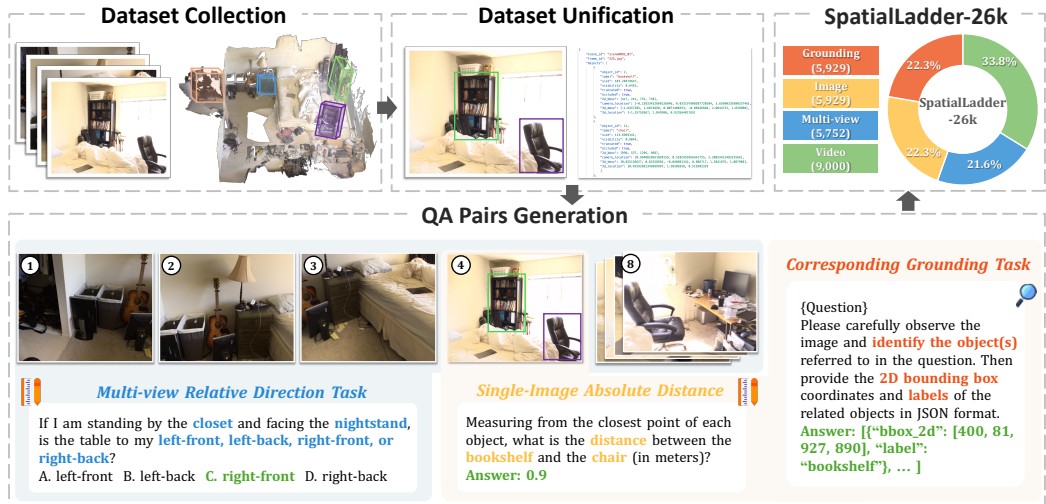

Figure 2: **Overview of SpatialLadder-26$k$ dataset construction pipeline** from raw data collection to question–answer pairs generation, with representative tasks including multi-view relative direction, single-image absolute distance, and corresponding grounding tasks.

## 3.1 DATASET CONSTRUCTION

Effective spatial reasoning requires diverse, high-quality training data spanning from basic perception to complex reasoning. We introduce SpatialLadder-26$k$, comprising 26,610 samples across four complementary task categories that form a complete spatial learning curriculum. Figure 2 illustrates our construction pipeline and dataset composition.

**Task Design and Hierarchy.** Our strategically designed dataset comprises four task categories: object localization (5,929 samples), single-image spatial reasoning (5,929 samples), multi-view spatial reasoning (5,752 samples), and video spatial reasoning (9,000 samples). Object localization establishes perceptual foundations via precise bounding box predictions for spatially-referenced objects. Spatial reasoning tasks span three modalities and seven dimensions: relative direction, relative distance, absolute distance, object size, counting, room size, and appearance order. Single-image tasks provide the entry point for static scene reasoning. Multi-view tasks require cross-perspective integration, synthesizing eight distinct viewpoints of identical environments. Video tasks incorporate temporal dynamics through 1–4 minute sequences at 24 fps, demanding coherent spatiotemporal understanding. This hierarchical progression ensures systematic capability development from foundational perception to complex spatiotemporal reasoning.

**Construction Pipeline.** Figure 2 details our standardized three-stage pipeline to ensure systematic data generation across all modalities. In the first stage, we collect ScanNet's (Dai et al., 2017) comprehensive 3D scene reconstructions for object localization, single-image spatial reasoning, and multi-view spatial reasoning, and carefully sample 9,000 videos from SR-91k (Ouyang et al., 2025) to support video spatial reasoning. In the second stage, we perform 3D-to-2D transformations and dataset unification, obtaining rich information including 3D bounding boxes, 2D bounding boxes, 3D absolute locations, 2D locations relative to the camera, visibility ratios and object sizes. In the third stage, we generate diverse question–answer pairs using templates adapted from VSI-Bench (Yang et al., 2025a) to construct tasks across different spatial reasoning scenarios. Further details on dataset construction (e.g. quality assurance, QA templates) are provided in B.1.

## 3.2 THREE-STAGE PROGRESSIVE TRAINING FRAMEWORK

Building upon SpatialLadder-26$k$, we design a training framework that systematically constructs spatial intelligence through three progressive stages, as illustrated in Figure 3, each addressing a specific level of the spatial reasoning hierarchy. The framework embodies the principle that robust

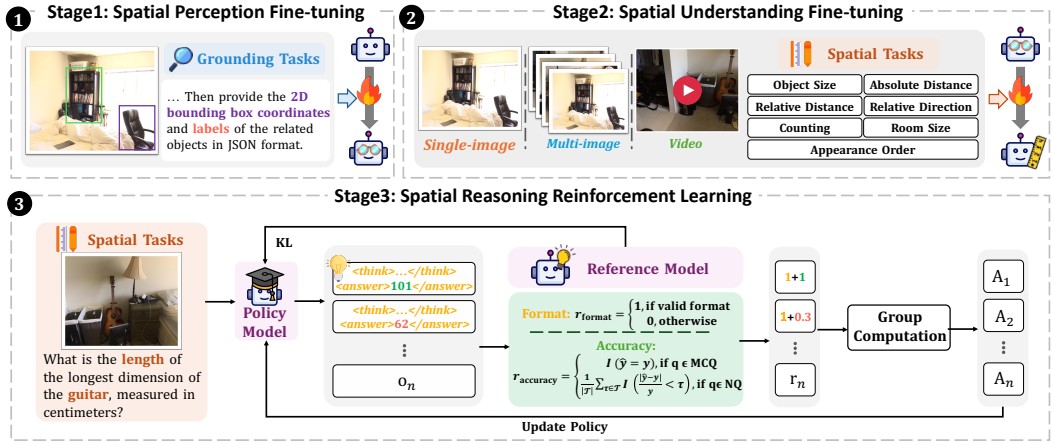

Figure 3: **Three-stage progressive training framework of SpatialLadder.** Stage 1 establishes perceptual grounding through object localization, Stage 2 develops spatial understanding across seven dimensions using multimodal tasks, and Stage 3 employs GRPO reinforcement learning with chain-of-thought generation to strengthen reasoning capabilities.

spatial reasoning emerges from the integration of perception, understanding, and reasoning, with each stage building upon foundations established in previous stages.

**Stage 1: Perceptual Grounding through Localization.** The first stage establishes foundational spatial perception via object localization on 6k SpatialLadder-$26k$ samples. The model learns to link visual inputs with spatial queries, producing JSON outputs containing object identities and 2D bounding boxes. This stage grounds abstract spatial concepts in concrete visual evidence. Through supervised fine-tuning, the model develops three core capabilities: distinguishing spatially relevant objects from background elements, robust detection tailored to spatial reasoning contexts, and mappings between linguistic descriptions and visual regions. The training emphasizes localization precision, as accurate object detection underpins all subsequent spatial reasoning. By focusing exclusively on perceptual tasks, we ensure strong visual grounding before advancing to complex reasoning.

**Stage 2: Spatial Understanding through Multi-dimensional Tasks.** The second stage broadens spatial comprehension by introducing comprehensive reasoning tasks that include size estimation, distance judgment, and orientation analysis across seven distinct spatial dimensions: relative direction, relative distance, absolute distance, object size, counting, room size, and appearance order. Training spans three modalities with distinct contributions: single-image tasks establish fundamental spatial relationships, multi-view tasks demand cross-perspective integration and implicit 3D understanding, while video tasks add temporal dynamics and motion tracking capabilities. This multimodal approach creates robust spatial representations that generalize across visual contexts. The supervised fine-tuning requires flexible adaptation between multiple-choice questions testing discrete concepts and numerical questions demanding precise measurements, developing comprehensive spatial understanding that transcends individual task types.

**Stage 3: Spatial Reasoning through Reinforcement Learning.** The final stage transforms spatial understanding into explicit reasoning capabilities through reinforcement learning with chain-of-thought (Wei et al., 2022; Xu et al., 2025) generation. We implement a carefully designed reward structure that evaluates both reasoning quality and answer correctness:

$$\mathcal{R}(o, y) = r_{\text{format}}(o) + r_{\text{accuracy}}(o, y) \tag{1}$$

Format rewards ensure structured reasoning by checking for proper `<think>` and `<answer>` tag usage, encouraging the model to explicitly articulate its reasoning process. Accuracy rewards are task-specific: binary for multiple-choice questions and graduated for numerical answers based on relative error thresholds. This dual reward structure prevents the model from generating plausible-sounding but incorrect reasoning chains.

We employ GRPO for stable policy optimization. For each question $q$, the model samples a series of candidate answers $\{o_1, o_2, ..., o_G\}$ from the policy model $\pi_{\text{old}}$ and optimizes the policy by maximizing the following objective function:

$$\mathcal{J}_{\text{GRPO}}(\theta) = \mathbb{E}_{q, o_i} \left[ \frac{1}{G} \sum_{i=1}^{G} \min \left( \frac{\pi_\theta(o_i|q)}{\pi_{\theta_{\text{old}}}(o_i|q)} A_i, \text{clip} \left( \frac{\pi_\theta(o_i|q)}{\pi_{\theta_{\text{old}}}(o_i|q)}, 1 \pm \varepsilon \right) A_i \right) - \beta \text{KL}[\pi_\theta \| \pi_{\text{ref}}] \right] \quad (2)$$

where $A_i = \frac{r_i - \text{mean}(r_1, r_2, ..., r_G)}{\text{std}(r_1, r_2, ..., r_G)}$ represents the advantage function computed through group-based calculation, $r_i$ denotes the reward value for answer $o_i$, $\text{KL}[\pi_\theta \| \pi_{\text{ref}}]$ represents the KL divergence (Kullback, 1951) between the policy model and reference model, and $\beta$ is the regularization hyperparameter.

## 4 EXPERIMENTS

### 4.1 EXPERIMENTAL SETUP

**Implementation Details.** We implement SpatialLadder using Qwen2.5-VL-3B (Bai et al., 2025) as the foundation model. The training procedure follows a three-stage progressive schedule with stage-specific hyperparameter configurations. Stages 1 and 2 employ supervised fine-tuning (Ouyang et al., 2022), while Stage 3 utilizes GRPO Guo et al. (2025) for reinforcement learning. Additional training details are provided in B.2.

**Evaluation Benchmarks.** We evaluate SpatialLadder on six benchmarks across in-domain and out-of-domain settings. For in-domain evaluation, we use VSI-Bench (Yang et al., 2025a) containing 5,155 video-based spatial reasoning questions and introduce two new benchmarks: SPBench-SI (1,009 single-image questions) and SPBench-MV (319 multi-view questions). Both SPBench benchmarks are constructed from ScanNet validation scenes using our pipeline, with strict scene-level separation ensuring zero overlap with training data. For out-of-domain evaluation, we assess generalization on CV-Bench (Tong et al., 2024) for 2D/3D vision tasks, SPAR-Bench (Zhang et al., 2025) for multi-difficulty spatial reasoning, ViewSpatial-Bench (Li et al., 2025a) for perspective-dependent spatial understanding, MMSI-Bench (Yang et al., 2025b) for diverse scenes spatial reasoning, and MindCube (Yin et al., 2025) for spatial mental modeling. Detailed benchmark and baseline descriptions are provided in Appendices C.1 and C.2, respectively.

### 4.2 MAIN RESULTS

**In-domain Performance.** Table 1 presents comprehensive evaluation on spatial reasoning benchmarks. SpatialLadder achieves state-of-the-art performance with 62.3% overall accuracy, surpassing all baselines including proprietary models. The performance gain is particularly pronounced on our proposed benchmarks: 70.2% on SPBench-SI (+29.9% over base model) and 70.9% on SPBench-MV (+34.3% over base model), demonstrating the effectiveness of our progressive training approach. Notably, while Spatial-MLLM achieves competitive performance on VSI-Bench (47.3%) using specialized 3D encoders, SpatialLadder attains comparable results of 45.7% (+16.3% over base model) using only the standard VLM architecture, validating that progressive training can substitute for architectural modifications. The consistent improvements across both numerical questions and multiple-choice questions indicate robust spatial understanding rather than task-specific overfitting. Further details of in-domain performance are presented in C.3.

**Generalization Analysis.** Table 2 demonstrates strong out-of-domain generalization with 45.0% overall accuracy, surpassing GPT-4o (42.7%) and achieving a 6.9% average improvement over the base model. The gains are consistent across diverse evaluation settings, confirming the robustness of our learned representations. Notably, the model achieves a significant +8.6% improvement on ViewSpatial-Bench, validating its proficiency in perspective-dependent spatial understanding and viewpoint transformation. Even more striking is the +10.2% gain on MindCube, which highlights the model's enhanced capacity for complex spatial mental modeling. Together with consistent improvements on CV-Bench (+3.1%), SPAR (+9.8%) and MMSI-Bench (+2.7%), these results demonstrate that our progressive training fosters generalized spatial intelligence that transfers effectively to novel viewpoints and complex cognitive tasks. Further details of out-of-domain performance are presented in C.4.

Table 1: **Evaluation Results on In-domain Benchmarks.** NQ and MCQ denotes numerical question and multiple-choice question, respectively. For each metric, **bold** numbers indicate the best performance, while underlined numbers represent the second-best performance.

| Model | VSI-Bench | | | SPBench-SI | | | SPBench-MV | | | Overall |
|---|---|---|---|---|---|---|---|---|---|---|
| | NQ | MCQ | Avg. | NQ | MCQ | Avg. | NQ | MCQ | Avg. | |
| *Proprietary Models* | | | | | | | | | | |
| GPT-4o (Hurst et al., 2024) | 33.4 | 34.6 | 34.0 | 24.5 | 60.3 | 42.4 | 40.7 | 59.4 | 48.2 | 41.5 |
| Gemini-2.0-Flash (Team et al., 2024) | 46.4 | **44.3** | 45.4 | 49.0 | 60.4 | 54.7 | 51.9 | 50.7 | 51.4 | 50.5 |
| *Open-Source Models* | | | | | | | | | | |
| InternVL-2.5-4B (Chen et al., 2024) | 30.6 | 34.1 | 32.4 | 31.8 | 53.3 | 42.5 | 37.7 | 51.4 | 43.2 | 39.4 |
| InternVL-2.5-8B (Chen et al., 2024) | 40.4 | 40.0 | 40.2 | 28.3 | 56.3 | 42.3 | 37.3 | 47.5 | 41.4 | 41.3 |
| Kimi-VL-A3B (Team et al., 2025) | 31.8 | 25.5 | 28.7 | 25.7 | 44.9 | 35.3 | 23.3 | 57.6 | 37.0 | 33.7 |
| LLaVA-OneVision-7B (Li et al., 2024a) | 34.5 | 31.2 | 32.9 | 25.4 | 41.0 | 33.2 | 20.6 | 49.6 | 32.2 | 32.8 |
| *Qwen2.5-VL-7B Based Spatial Models* | | | | | | | | | | |
| Qwen2.5-VL-7B (Bai et al., 2025) | 37.1 | 34.6 | 35.8 | 36.3 | 60.5 | 48.4 | 28.9 | 49.8 | 37.3 | 40.5 |
| SpaceR-7B (Ouyang et al., 2025) | 47.8 | 41.2 | 44.5 | 35.7 | 61.5 | 48.6 | 63.2 | 53.7 | 59.4 | 50.8 |
| VILASR-7B (Wu et al., 2025c) | 47.4 | 43.4 | 45.4 | 36.6 | 63.7 | 50.2 | 56.2 | 59.6 | 57.6 | 51.1 |
| Video-R1 (Feng et al., 2025) | 33.8 | 32.9 | 33.4 | 27.7 | 62.0 | 44.8 | 32.5 | 53.0 | 40.7 | 39.6 |
| *Qwen2.5-VL-3B Based Spatial Models* | | | | | | | | | | |
| Qwen2.5-VL-3B (Bai et al., 2025) | 26.0 | 33.0 | 29.4 | 24.3 | 56.2 | 40.3 | 25.6 | 53.2 | 36.6 | 35.4 |
| Spatial-MLLM-4B (Wu et al., 2025a) | **51.5** | 43.1 | **47.3** | 38.1 | 49.3 | 43.7 | 63.7 | 58.9 | 61.8 | 50.9 |
| SpatialLadder-3B | 50.8 | 40.5 | 45.7 | **58.6** | **81.8** | **70.2** | **68.2** | **75.0** | **70.9** | **62.3** |
| *Improvement* | +24.9 | +7.6 | +16.3 | +34.3 | +25.6 | +29.9 | +42.6 | +21.8 | +34.3 | +23.4 |

Table 2: **Evaluation Results on Out-of-domain Benchmarks.** For each benchmark, **bold** numbers indicate the best performance, while underlined numbers represent the second-best performance.

| Model | CV-Bench | SPAR-Bench | ViewSpatial | MMSI-Bench | MindCube | Overall |
|---|---|---|---|---|---|---|
| GPT-4o (Hurst et al., 2024) | 75.4 | **36.4** | 32.6 | **30.3** | 38.8 | 42.7 |
| InternVL-2.5-4B (Chen et al., 2024) | 74.4 | 30.6 | 37.9 | 26.3 | 18.3 | 37.6 |
| InternVL-2.5-8B (Chen et al., 2024) | 76.5 | 36.3 | 43.2 | 25.7 | 18.7 | 40.1 |
| LLaVA-OneVision-7B (Li et al., 2024a) | 58.3 | 31.2 | 27.5 | 24.5 | **47.3** | 37.8 |
| Qwen2.5-VL-7B (Bai et al., 2025) | **79.0** | 30.2 | 37.9 | 25.9 | 29.3 | 40.5 |
| Qwen2.5-VL-3B (Bai et al., 2025) | 70.6 | 24.6 | 35.6 | 26.5 | 33.2 | 38.1 |
| SpatialLadder-3B | 73.7 | 34.4 | **44.2** | 29.2 | 43.4 | **45.0** |
| *Improvement* | +3.1 | +9.8 | +8.6 | +2.7 | +10.2 | +6.9 |

## 4.3 ABLATION STUDIES

**Component Analysis.** Figure 5 reveals the critical interdependence of SpatialLadder's components. Stage 2 (spatial understanding fine-tuning) proves most essential, with its removal causing a 9.4% accuracy drop, validating explicit spatial cognition as the training cornerstone. Stages 1 and 3 contribute meaningfully (1.8% and 2.1% drops respectively), confirming progressive training's value. Excluding single-image and multi-view data causes the most severe degradation (16.4% loss), affecting not only corresponding benchmarks but also video-based VSI-Bench performance. This demonstrates that multimodal diversity is fundamental for robust spatial reasoning across all modalities. Chain-of-thought reasoning provides consistent 0.8% gains, validating explicit reasoning in spatial tasks.

**Training Dynamics.** Figure 4 demonstrates that the complete SpatialLadder-3B consistently outperforms variants missing Stage 1 or Stage 2 across accuracy reward curves. The reward standard deviation analysis reveals superior training stability for the full model, exhibiting the most significant variance reduction and smoothest convergence patterns. On VSI-Bench evaluation, the complete framework achieves highest accuracy while ablated variants show notable degradation, with Stage 2's absence producing the most pronounced performance decline. Appendix D.1 provides additional training dynamics with and without chain-of-thought.

## 4.4 IN-DEPTH ANALYSIS

**Semantic Consistency Emerges through Reinforcement Optimization.** We employ semantic entropy (Kuhn et al., 2023) to quantify model uncertainty. As shown in Figure 6, during Stages 1-2 where the model establishes perceptual foundations and spatial understanding capabilities, entropy increases from 1.24 to 1.47 as spatial capabilities transcend initial misconceptions and expand the

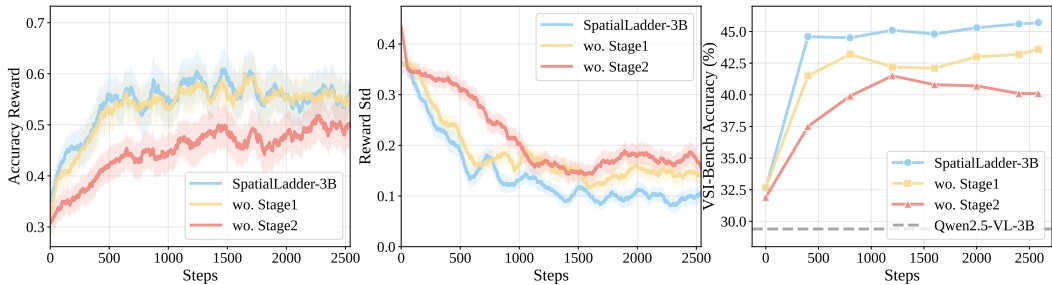

Figure 4: **Impact of progressive training stages.** Left: accuracy rewards over training steps; Middle: reward standard deviation over training steps; Right: VSI-Bench performance comparison.

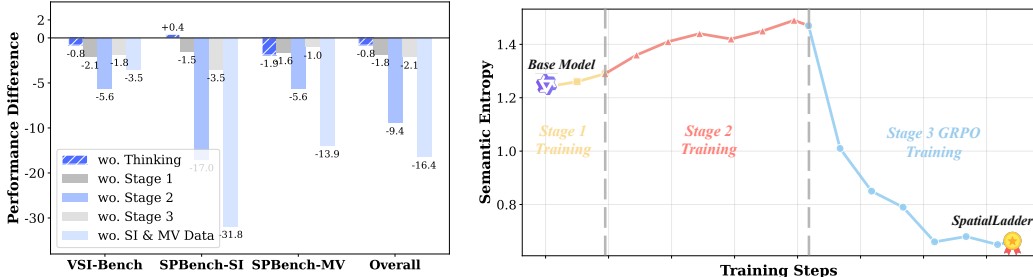

Figure 5: Ablation study results.

Figure 6: Semantic entropy dynamics.

exploration space for improved reasoning. Subsequently, during Stage 3 reinforcement learning, entropy steadily declines from 1.47 to 0.66, marking the transition from broad exploration to focused reasoning convergence. This quantitative progression validates our three-stage strategy: establishing comprehensive foundations, expanding the reasoning space, and achieving robust spatial intelligence through convergence. Further details are provided in Appendix C.6.

**Visual Attention Becomes Precisely Object-centric through Progressive Training.** To understand how our training framework influences internal mechanisms, we analyzed the visual attention patterns of SpatialLadder and Qwen2.5-VL-3B. Qualitatively, Figure 7 (top) reveals that SpatialLadder exhibits significantly more concentrated attention on task-relevant objects. Crucially, in relational tasks, the model generates distinct, simultaneous attention hotspots for all involved entities, whereas the base model typically exhibits diffuse or singular attention patterns.

We conducted a quantitative evaluation of attention distributions using 400 samples from SPBench-SI, with two metrics: Visual Attention IoU, which measures the concentration of attention within object bounding boxes, and Visual Attention Entropy, which quantifies the degree of attention dispersion across the visual field. As shown in Figure 7 (bottom), SpatialLadder achieves superior overall performance with 73.5% accuracy and 37.7% visual attention IoU compared to the base model's 32.1% accuracy and 33.8% IoU. Additionally, SpatialLadder exhibits lower visual attention entropy (0.176 vs. 0.193), confirming that our progressive training effectively reshapes the model's mechanism to precisely focus on relevant targets during spatial reasoning. Notably, this phenomenon is consistent across all four task types—spanning both single-object and multi-object scenarios—where SpatialLadder consistently demonstrates higher accuracy aligned with superior attention concentration. Further details are provided in Appendix C.7.

**Hierarchical Reasoning Structures Develop Naturally from Perceptual Foundations.** Qualitative analysis indicates that SpatialLadder acquires systematic spatial cognition through foundational perceptual training. As illustrated in Figure 8, the model exhibits a hierarchical cognitive architecture in which accurate identification of spatial elements serves as the perceptual basis for constructing coherent logical reasoning chains via structured analysis. Beyond basic reasoning, the model demonstrates advanced metacognitive capabilities, including self-verification and error correction, which help maintain consistency throughout the reasoning process.

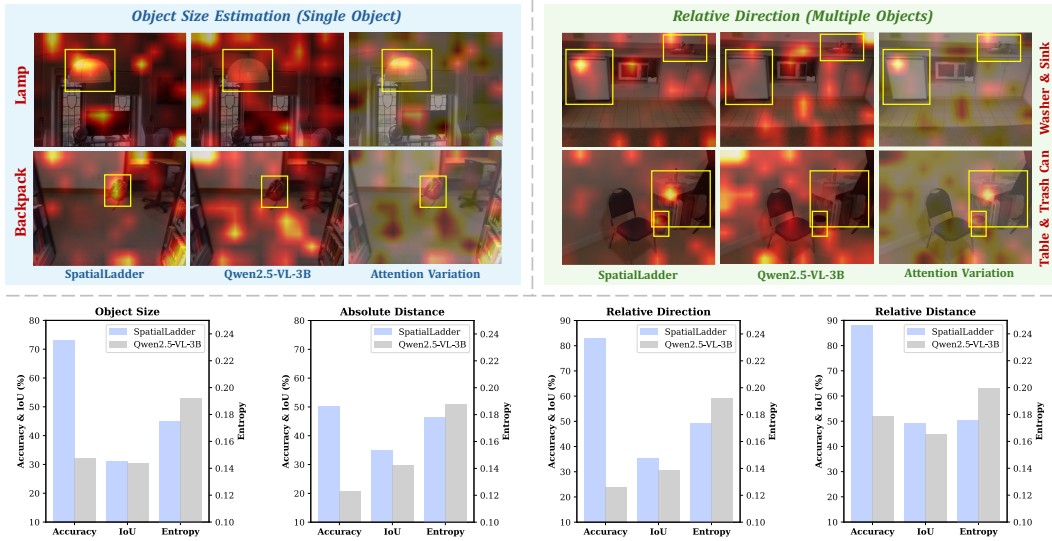

Figure 7: **Visual attention comparison between SpatialLadder and Qwen2.5-VL-3B.** Top: Representative attention distribution patterns for both models. Bottom: Quantitative analysis of performance accuracy, attention IoU, and attention entropy metrics.

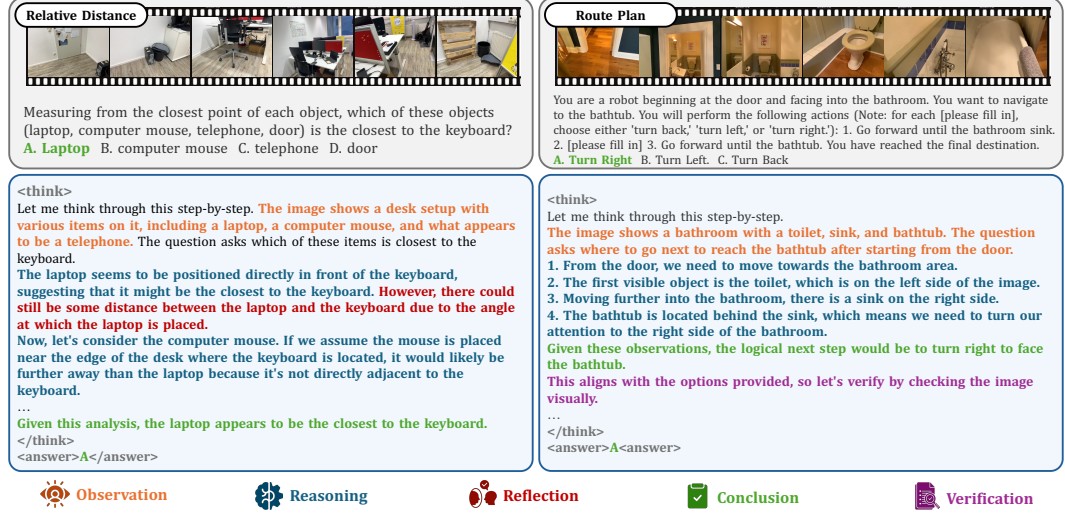

Figure 8: Hierarchical reasoning process demonstration in SpatialLadder.

In relative distance tasks, SpatialLadder explicitly decomposes spatial relationships into interpretable subcomponents, while in path-planning scenarios it systematically evaluates complex spatial layouts and navigational constraints. This structured chain-of-thought behavior highlights how robust perceptual grounding naturally scaffolds higher-order spatial reasoning.

Importantly, SpatialLadder consistently produces correct conclusions accompanied by clear and well-organized reasoning steps. These results suggest that strengthening foundational spatial perception effectively supports hierarchical reasoning, thereby validating the design and effectiveness of our training framework.

**Reinforcement Learning Unlocks Performance Beyond the Limits of Supervised Training.** To isolate the unique contribution of Stage3, we perform a controlled comparison between extended supervised fine-tuning and our RL stage. Using the standard Stage1–2 pipeline as the baseline, we investigate whether adding an extra epoch of Stage-2 SFT can replicate the gains brought by RL.

Table 3: **Performance comparison between extended SFT and RL.** RL surpasses the performance limits of SFT, while extended SFT leads to performance degradation.

| Model | VSI-Bench | SPBench-SI | SPBench-MV | Overall |
|---|---|---|---|---|
| Qwen2.5-VL-3B (Backbone) | 29.4 | 40.3 | 36.6 | 38.8 |
| *+ Stage 1-2 SFT Training* | 43.9 | 68.5 | 69.9 | 60.2 |
| *+ Stage 1-2 (2 Epochs) SFT Training* | 43.4 | 64.3 | 67.0 | 58.2 (↓ **2.0**) |
| *+ Stage 1-3 Training (Ours)* | **45.7** | **70.2** | **70.9** | **62.3** (↑ **2.1**) |

As shown in Table 3, when trained on the same dataset for the same number of additional epochs, the model with extended SFT consistently underperforms the baseline across all benchmarks, resulting in an overall performance drop of 2.0%. This degradation suggests that prolonged supervised training leads to overfitting and negatively impacts generalization.

In contrast, incorporating Stage3 (RL) effectively avoids this performance collapse and further surpasses the SFT performance ceiling. The RL-enhanced model achieves consistent improvements across all benchmarks, yielding an overall gain of 2.1%. These results demonstrate that reinforcement learning plays a critical role in enhancing spatial reasoning capability, enabling performance improvements that cannot be achieved through purely supervised training alone.

**Well Preservation of General Multimodal Capabilities.** Since SpatialLadder-26$k$ consists exclusively of spatial perception and spatial reasoning tasks, a natural concern is whether our three-stage training pipeline induces catastrophic forgetting of general multimodal capabilities. To assess this risk, we evaluate the model on two widely used general-purpose multimodal benchmarks: MMBench(Liu et al., 2024), which measures broad vision–language understanding, and MMMU(Yue et al., 2024), which focuses on multidisciplinary visual reasoning.

Table 4: Results on general benchmarks

| Method | MMBench | MMMU |
|---|---|---|
| Qwen2.5-VL-3B | 83.3 | 48.3 |
| SpatialLadder-3B | 82.4 | 47.1 |
| Δ | -0.9 | -1.2 |

As reported in Table 4, SpatialLadder shows only marginal performance reductions compared to the base model, with drops of 0.8% on MMBench and 1.2% on MMMU. Given the substantial gains achieved on spatial reasoning benchmarks, these minor degradations indicate that our progressive training strategy significantly enhances spatial intelligence while largely preserving the model's general multimodal competence.

Additional analysis about comparison with other spatial reasoning dataset, dataset scaling and progressive training order are provided in D.2, D.3 and D.4 respectively.

## 5 CONCLUSION

This work addresses the perception–reasoning gap in VLMs for spatial tasks through a systematic solution. We introduce SpatialLadder-26$k$, a multimodal dataset covering object localization as well as single-view, multi-view, and video-based spatial reasoning, and propose a three-stage progressive training framework that develops spatial intelligence from perception to understanding and reasoning. Built upon this framework, our SpatialLadder model achieves state-of-the-art performance across multiple benchmarks, demonstrating strong generalization both in-domain and out-of-domain. Extensive ablation studies further validate the effectiveness of each component. Together, these results establish a new paradigm for spatial reasoning in VLMs and open up promising directions for future research, as discussed in Appendix E.

## 6 ACKNOWLEDGEMENTS

This work was supported by National Natural Science Foundation of China (No. 62506332) and the Key Research and Development Program of Zhejiang Province, China (No. 2024C03255).

ETHICS STATEMENT

This work does not involve human subjects, personal data, or sensitive information. All datasets used in our experiments (VSI-Bench, SPBench-SI, SPBench-MV, CV-Bench, SPAR-Bench, ViewSpatial-Bench) are publicly available benchmark datasets designed for evaluating visual spatial reasoning in VLMs. We strictly adhered to ethical research practices and did not conduct any data collection that could raise privacy, security, or fairness concerns. Our methods—SpatialLadder-26$k$ dataset and progressive three-stage training framework—address the perception-reasoning gap in VLMs for spatial tasks, developing robust spatial reasoning capabilities without introducing risks of harmful applications. To the best of our knowledge, this research complies with the ICLR Code of Ethics and poses no foreseeable ethical concerns.

REPRODUCIBILITY STATEMENT

We have made extensive efforts to ensure the reproducibility of our work. Comprehensive details of dataset construction are provided in B.1, while training configurations and hyperparameters are systematically reported in B.2. Detailed dataset descriptions are documented in C.1. The comprehensive evaluation results are outlined in C.3 and C.4, and the implementation details of our analysis experiments are thoroughly described in C.6 and C.7. Upon acceptance, we will release our models, together with training and evaluation code, to facilitate replication and further research.

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

TECHNICAL APPENDICES AND SUPPLEMENTARY MATERIAL

## A    PRELIMINARY ANALYSIS

To validate our hypothesis that spatial reasoning failures stem from inadequate perceptual grounding rather than reasoning incapacity, we conducted controlled experiments examining how progressive perceptual hints affect model performance. We constructed a diagnostic dataset of 200 spatial orientation tasks from ScanNet validation scenes, each requiring determination of relative positions between object pairs from the camera's perspective.

We evaluated Qwen2.5-VL-3B (Bai et al., 2025) under three conditions: (1) baseline with raw images and questions, (2) location hints adding colored bounding boxes around target objects, and (3) full hints incorporating directional arrows within bounding boxes. Results demonstrate monotonic improvement with enhanced perceptual grounding: baseline accuracy of 36.5% improves to 41.5% with location hints (+5.0%) and 46.0% with directional hints (+4.5% additional).

These findings directly motivate our progressive training approach. This experimental evidence establishes that robust spatial reasoning cannot be achieved through end-to-end learning but requires systematic construction from perceptual foundations to abstract reasoning. After training, our model SpatialLadder achieves consistently high performance across all conditions: 82.0% without hints, 82.5% with location hints, and 83.5% with full hints. This minimal variation (1.5% range) demonstrates that progressive training successfully internalizes spatial perception capabilities, eliminating dependence on external scaffolding.

## B    ADDITIONAL METHOD DETAILS

### B.1    DETAILS OF SPATIALLADDER-26$k$ CONSTRUCTION

Table 5: Question templates for tasks in SpatialLadder-26$k$.

| Task | Question Template |
|---|---|
| Object Counting | *How many {**category**}(s) appear?* |
| Absolute Distance | *Measuring from the closest point of each object, what is the distance between the {**object 1**} and the {**object2**} (in meters)?* |
| Object Size | *What is the length of the longest dimension (length, width, or height) of the {**object**}, measured in centimeters?* |
| Relative Distance | *Measuring from the closest point of each object, which of these two objects ({**choice a**}, {**choice b**}) is closer to the {**category**}?* |
| Relative Direction | The question template for relative direction tasks varies between single-image and multi-view modalities. 

 • **Single-image**: *From the camera's perspective, is the {**object 1**} to the {**object 2**}'s {**choice a**}, {**choice b**}, {**choice c**} or {**choice d**}?* 

 • **Multi-view**: *If I am standing by the {**positioning object**} and facing the {**orienting object**}, is the {**querying object**} to my {**choice a**}, {**choice b**}, {**choice c**} or {**choice d**}? The directions refer to the quadrants of a Cartesian plane (if I am standing at the origin and facing along the positive y-axis).?* |
| Object Localization | *{**question**} Please carefully observe the image first to identify the object(s) referred to in the question. Note that each object type appears only once in the image. Then provide the 2D bounding box coordinates and labels of the related objects in JSON format.* |

**Question-answer Generation Details**    Based on the metadata unified from ScanNet, we further construct question–answer pairs for various single-image and multi-view spatial reasoning tasks. The generation process is described as follows:

- **Object Counting**: This task involves a single object category and is designed exclusively for the multi-view setting. We first identify the target object category, then determine the

number of distinct instances that appear across all views by checking their presence in each camera view, and finally use the aggregated count as the answer.

- **Absolute Distance**: This task involves two objects. Using their 3D locations from the metadata, we calculate the Euclidean distance between them as the answer. To ensure clarity and avoid ambiguity, we enforce that the computed distance must exceed the minimum size of the two objects.

- **Object Size**: This task focuses on a single object. We estimate the object's size using the maximum dimension of its 3D bounding box, while filtering out objects that are either excessively large or small to ensure human-scale spatial reasoning.

- **Relative Distance**: This task involves three objects—a target object and two candidate objects. We compare the distances from the target object to each candidate and select the closer one as the answer. To prevent ambiguous cases, we require the larger distance to be at least twice the smaller one.

- **Relative Direction**: The construction of this task differs between single-image and multi-view modalities. For the single-image setting, the task involves two objects, and their relative orientation is defined with respect to the camera viewpoint. Specifically, we compute the left/right relation based on their 2D locations relative to the image plane, and the front/back relation based on their depth from the camera. Composite relations (e.g., left-front, left-back, right-front, right-back) are included when applicable. For the multi-view setting, the task involves three objects: a positioning object, an orienting object, and a querying object. We define vectors from the positioning object to the orienting object ($\vec{a}$) and to the querying object ($\vec{b}$), and compute the angle $\theta$ between them using $\cos(\theta) = \frac{\vec{a} \cdot \vec{b}}{|\vec{a}||\vec{b}|}$. The relative direction of the querying object is then determined according to this angular relationship.

The question templates employed for generating QA pairs across various spatial reasoning and object localization tasks in SpatialLadder-26$k$ are detailed in Table 5.

**Quality Assurance.** To ensure dataset reliability, we implement multiple filtering mechanisms. Scene diversity is maintained by limiting questions per scene to prevent overfitting to specific environments. Object diversity is enforced by restricting the number of samples constructed from the same object type within a scene, avoiding bias toward particular categories and ensuring question variety. Noisy objects (e.g. wall, floor and ceiling) are filtered to focus on human-scale spatial reasoning. Minimum visibility threshold (40%) ensures that spatial judgments are based on sufficient visual evidence. Objects must be uniquely identifiable within their context to avoid ambiguity. These constraints eliminate approximately 90% of initially generated samples, resulting in a high-quality dataset where each sample provides clear spatial learning signals.

Table 6: Detailed statistics of the spatial reasoning subset in SpatialLadder-26$k$.

| Modality | Numerical Question | | | | Multiple-choice Question | | | Total |
|---|---|---|---|---|---|---|---|---|
| | Obj. Cnt. | Abs. Dist. | Obj. Size | Room Size | Rel. Dist. | Rel. Dir. | Appr. Order | |
| Single-Image | - | 1,127 | 1,514 | - | 1,034 | 2,253 | - | 5,929 |
| Multi-View | 217 | 817 | 1,867 | - | 635 | 2,162 | - | 5,752 |
| Video | 507 | 1,500 | 1,331 | 150 | 1,134 | 3,061 | 1,317 | 9,000 |

**The Statics of SpatialLadder-26$k$.** The distribution of spatial reasoning tasks across different modalities in our constructed SpatialLadder-26$k$ is presented in Table 6. In SpatialLadder-26$k$, object localization tasks are based on the single-image modality but remain independent of specific spatial reasoning task types. Each single-image spatial reasoning task is paired with a corresponding object localization task.

### B.2 DETAILS OF TRAINING IMPLEMENTATION

**Prompt Used for Training.** The system prompt and user prompt employed in the SpatialLadder three-stage training framework are presented in the boxes below. The post prompt design in Stage 2 and Stage 3 varies across task types: for multiple-choice questions, the prompt guides the model to

output the corresponding option, whereas for numerical questions, the prompt instructs the model to provide a numerical answer.

---

**Prompt for Stage 1**

**System Prompt**: *"You are a helpful assistant."*
**User Prompt**: **{question}** + *"Please carefully observe the image first to identify the object(s) referred to in the question. Note that each object type appears only once in the image. Then provide the 2D bounding box coordinates and labels of the related objects in JSON format."*

---

**Prompt for Stage 2**

**System Prompt**: *"You are a helpful assistant."*
**User Prompt**: **{question}** + **Post Prompt["question type"]**
**Post Prompt**:

- Multiple-choice Question: *"Please answer with the option's letter from the given choices (e.g., A, B, etc.) directly."*

- Numerical Question: *"Please answer the question using a numerical value (e.g., 42 or 3.1) directly."*

---

**Prompt for Stage 3**

**System Prompt**: *"You are a helpful assistant."*
**User Prompt**: **{question}** + *"Please think about this question as if you were a human pondering deeply. Engage in an internal dialogue using expressions such as 'let me think', 'wait', 'Hmm', 'oh, I see', 'let's break it down', etc, or other natural language thought expressions. It's encouraged to include self-reflection or verification in the reasoning process."* + **Post Prompt["question type"]**
**Post Prompt**:

- Multiple-choice Question: *"Please provide your detailed reasoning between the* `<think>` `</think>` *tags, and then answer the question with the option's letter from the given choices (e.g., A, B, etc.) within the* `<answer>` `</answer>` *tags.""*

- Numerical Question: *"Please provide your detailed reasoning between the* `<think>` `</think>` *tags, and then answer the question with a numerical value (e.g., 42 or 3.1) within the* `<answer>` `</answer>` *tags."*

---

Table 7: Hyperparameter used in Stage 1-2.

| Hyperparameter | Value |
| --- | --- |
| per_device_train_batch_size | 1 |
| gradient_accumulation_steps | 8 |
| bf16 | true |
| data_seed | 42 |
| gradient_checkpointing | true |
| attn_implementation | flash_attention_2 |
| lr_scheduler_type | cosine |
| warmup_ratio | 0.1 |
| num_train_epochs | 1 |
| max_pixels | 100,352 |
| min_pixels | 12,544 |

Table 8: Hyperparameter used in Stage 3.

| Hyperparameter | Value |
| --- | --- |
| num_generations | 8 |
| per_device_train_batch_size | 2 |
| gradient_accumulation_steps | 4 |
| bf16 | true |
| data_seed | 42 |
| gradient_checkpointing | true |
| attn_implementation | flash_attention_2 |
| num_train_epochs | 1 |
| max_pixels | 100,352 |
| min_pixels | 12,544 |
| $\beta$ | 0.01 |

**Reproduction details.** Our model was trained on a 4 × NVIDIA A6000 GPU cluster with 48GB memory per device. The training process consisted of three distinct stages: stages 1-2 employed supervised fine-tuning methodology implemented via the HuggingFace Transformers Reinforcement Learning (TRL) framework, with corresponding hyperparameters detailed in Table 7. Stage 3 utilized GRPO reinforcement learning, implemented through the VLM-R1 framework (Shen et al., 2025), with corresponding hyperparameters specified in Table 8.

**Details of Stage 1 and Stage 2.** In our progressive three-stage training framework, Stage 1 develops spatial perception capabilities through targeted spatial localization tasks, while Stage 2

enhances spatial understanding through multi-dimensional spatial reasoning tasks. Both stages employ supervised fine-tuning methodology, optimizing the standard cross-entropy loss function:

$$\mathcal{L}_{\text{ce}}(\theta) = -\sum_i \log P\left(o^{(i)} \mid o^{(1:i-1)}, q, v\right) \tag{3}$$

where $v$ represents the input visual information, $q$ denotes the textual query and instruction, $o^{(i)}$ represents the $i$-th token in the generated response, and $o^{(1:i-1)}$ corresponds to the preceding context tokens. This supervised learning approach establishes the foundational spatial capabilities that are subsequently refined through reinforcement learning in Stage 3.

**Details of Cold Start.** Before the formal GRPO training in Stage 3, we perform a cold-start (Guo et al., 2025) phase to ensure that the model can more reliably generate outputs that satisfy the required format. Specifically, we adopt a rejection sampling strategy to construct chain-of-thought augmented data with composite formatting constraints. Concretely, based on the spatial reasoning tasks from SpatialLadder-26$k$, we use the Qwen2.5-VL-7B model to generate candidate question–answer pairs with reasoning chains. The generated responses are then filtered using a reward function under two criteria: (1) the response must strictly satisfy the predefined formatting requirements, and (2) its accuracy reward must exceed a predefined threshold. The resulting rejection-sampled dataset is defined as:

$$\mathcal{D}_{\text{coldstart}} = \{(v_i, q_i, o_i) \mid (v_i, q_i, o_i, y_i) \in \mathcal{D}_{\text{candicate}} \wedge \mathcal{R}(o_i, y_i) > 1 + \lambda\} \tag{4}$$

where $\mathcal{D}_{\text{candicate}}$ denotes the set of candidate question–answer pairs generated by Qwen2.5-VL-7B on SpatialLadder-26$k$, $v_i$ represents the visual input of the $i$-th question, $q_i$ denotes the question text, $o_i$ corresponds to the model's response for the $i$-th question, and $\lambda$ is the accuracy reward threshold. This process yielded a total of 1,255 cold-start training samples.

**Details of Reward Function** Stage 3 introduces the GRPO reinforcement learning algorithm to further stimulate the model's spatial reasoning capabilities through carefully designed reward mechanisms. Our reward system includes format rewards and accuracy rewards.

The format reward ensures structured model outputs by requiring the model to place its reasoning process and final answer within `<think>` ... `</think>` and `<answer>` ... `</answer>` tags, respectively:

$$r_{\text{format}}(o) = \begin{cases} 1, & \text{if } o \text{ matches format} \\ 0, & \text{otherwise} \end{cases} \tag{5}$$

The accuracy reward employs differentiated evaluation strategies based on question types. For multiple-choice questions, we adopt a strict exact matching criterion:

$$r_{\text{mc}}(o, y) = \mathbb{I}(o = y) \tag{3}$$

where $o$ represents the model's prediction and $y$ denotes the ground truth label for the question.

For numerical answer questions, considering the continuous nature of numerical predictions, we design a weighted relative accuracy measure based on confidence intervals:

$$r_{\text{num}}(o, y) = \frac{1}{|\mathcal{T}|} \sum_{\tau \in \mathcal{T}} \mathbb{I}\left(\frac{|o - y|}{y} < \tau\right) \tag{6}$$

where $\mathcal{T} = [0.50, 0.55, ..., 0.95]$ represents a series of confidence thresholds.

The unified accuracy reward function is defined as:

$$r_{\text{accuracy}}(o, y) = \begin{cases} r_{\text{mc}}(o, y), & \text{if } q \in \text{MCQ} \\ r_{\text{num}}(o, y), & \text{if } q \in \text{NQ} \end{cases} \tag{7}$$

where $q$ represents the input question, MCQ denotes the set of multiple-choice questions, NQ denotes the set of numerical answer questions.

The final reward function integrates both format and accuracy dimensions:

$$\mathcal{R}(o, y) = r_{\text{format}}(o) + r_{\text{accuray}}(o, y) \tag{8}$$

## C  ADDITIONAL EXPERIMENTS DETAILS

Table 9: Detailed statistics of the SPBench-SI and SPBench-MV.

| Benchmark | Numerical Question | | | Multiple-choice Question | | Total |
|---|---|---|---|---|---|---|
| | Obj. Cnt. | Abs. Dist. | Obj. Size | Rel. Dist. | Rel. Dir. | |
| SPBench-SI | - | 149 | 463 | 91 | 306 | 1,009 |
| SPBench-MV | 70 | 30 | 158 | 17 | 44 | 319 |

### C.1  DETAILS OF BENCHMARKS

- **VSI-Bench** (Yang et al., 2025a): VSI-Bench is a comprehensive evaluation benchmark for assessing visual-spatial intelligence in Multimodal Large Language Models (MLLMs) through egocentric video understanding. The benchmark comprises over 5,000 question-answer pairs from 288 real-world videos sourced from ScanNet Dai et al. (2017), ScanNet++ Yeshwanth et al. (2023), and ARKitScenes (Baruch et al., 2021), spanning diverse environments across multiple geographic regions.

- **SPBench-SI & SPBench-MV**: SPBench-SI and SPBench-MV are evaluation benchmarks constructed using the SpatialLadder-$26k$ pipeline applied to the ScanNet validation set. SPBench-SI serves as a single-image evaluation benchmark designed to assess models' spatial understanding and reasoning capabilities from individual viewpoints, encompassing four task categories: absolute distance, object size, relative distance, and relative direction, with a total of 1,009 samples. SPBench-MV constitutes a multi-view evaluation benchmark that requires models to perform joint spatial modeling across multiple viewpoints. SPBench-MV additionally incorporates object counting tasks to evaluate models' capabilities in identifying and enumerating objects within multi-view scenarios. Both benchmarks undergo rigorous quality control through the standard pipeline filtering strategies supplemented by manual curation to ensure data disambiguation and high-quality annotations. The detailed statistics of SPBench-SI and SPBench-MV are provided in Table 9.

- **CV-Bench** (Tong et al., 2024): CV-Bench addresses limitations of existing vision-centric benchmarks through 2,638 manually-inspected examples. The benchmark repurposes established vision datasets—ADE20k (Zhou et al., 2017), COCO (Lin et al., 2014), and OMNI3D (Brazil et al., 2023)—to evaluate MLLMs on fundamental computer vision tasks. The evaluation encompasses 2D spatial comprehension through spatial relationships and object counting, while 3D understanding is assessed via depth ordering and relative distance estimation.

- **SPAR-Bench** (Zhang et al., 2025): SPAR-Bench constitutes a comprehensive evaluation framework for systematically assessing spatial perception and reasoning capabilities in VLMs. The benchmark encompasses 20 diverse spatial understanding tasks spanning single-view, multi-view, and temporal video modalities, incorporating 7,207 manually verified question-answer pairs to ensure annotation quality and reliability.

- **ViewSpatial-Bench** (Li et al., 2025a): ViewSpatial-Bench is a comprehensive evaluation framework comprising over 5,700 question-answer pairs across 1,000+ 3D scenes from

ScanNet (Dai et al., 2017) and MS-COCO (Lin et al., 2014) validation datasets. This benchmark evaluates VLMs' spatial localization capabilities from both egocentric and allocentric viewpoints, addressing the critical gap in perspective-taking abilities essential for embodied (Wang et al., 2025b) interaction and multi-agent collaboration.

- **MMSI-Bench** (Yang et al., 2025b): MMSI-Bench focuses on multi-image spatial intelligence for MLLMs. The benchmark contains 1,000 manually-curated multiple-choice questions derived from over 120,000 images, each paired with carefully designed distractors and a stepwise reasoning process. MMSI-Bench evaluates models on core spatial reasoning skills, including grounding, overlap matching, scene reconstruction, situation transformation, and spatial logic.

- **MindCube** (Yin et al., 2025): MindCube is a benchmark for evaluating spatial reasoning in VLMs from limited visual inputs. It contains 21,154 questions across 3,268 images, testing core capabilities such as cognitive mapping, perspective-taking, and mental simulation. MindCube is designed to identify gaps in current VLMs' spatial understanding and support research on structured spatial representations.

## C.2 DETAILS OF BASELINES

- **GPT-4o** (Hurst et al., 2024): GPT-4o is a multilingual and multimodal generative transformer released in May 2024, supporting text, image, and audio understanding and generation with strong general capabilities.

- **Gemini-2.0-Flash** (Team et al., 2024): Gemini 2.0 Flash is a multimodal model optimized for agent-centric applications, featuring efficient computation, integrated tool use, multi-modal generation, and a 1M-token context window with improved quality over previous Flash versions.

- **InternVL-2.5-4B/8B** (Chen et al., 2024): InternVL 2.5 is an enhanced version of InternVL 2.0 with improved training strategies and data quality, achieving competitive performance across reasoning, document understanding, and video comprehension.

- **Kimi-VL-A3B** (Team et al., 2025): Kimi-VL-A3B is an efficient MoE-based VLM (activating 2.8B parameters) with strong multimodal reasoning, long-context processing, and agent capabilities.

- **LLaVA-OneVision-7B** (Li et al., 2024a): LLaVA-OneVision-7B is an open MLLM performing well on single-image, multi-image, and video tasks, showing strong cross-modal transfer and particularly effective video understanding.

- **Qwen2.5-VL-3B/7B** (Bai et al., 2025): Qwen2.5-VL is a VLM with enhanced recognition, localization, document parsing, and long-video comprehension, supported by dynamic resolution handling for variable-sized inputs.

- **SpaceR-7B** (Ouyang et al., 2025): SpaceR-7B is a video spatial reasoning model trained with reinforcement learning using verifiable rewards. It incorporates a map imagination mechanism to infer spatial layouts during reasoning and demonstrates strong performance on VSI-Bench, surpassing GPT-4o.

- **VILASR-7B** (Wu et al., 2025c): VILASR-7B introduces a "drawing-to-reason" paradigm, enabling the model to perform spatial reasoning through elementary drawing operations. It uses a three-stage training pipeline (synthetic cold-start, reflective rejection sampling, and reinforcement learning) to learn structured spatial reasoning and achieves strong results on spatial benchmarks.

- **Video-R1** (Feng et al., 2025): Video-R1 extends the R1 reasoning paradigm to video understanding, leveraging the T-GRPO algorithm to better capture temporal dynamics. The model is trained on both image and video reasoning tasks and demonstrates robust spatial-temporal reasoning, surpassing GPT-4o on VSI-Bench while performing well on general video benchmarks.

- **Spatial-MLLM-4B** (Wu et al., 2025a): Spatial-MLLM-4B is a dual-encoder spatial reasoning framework, combining a pretrained 2D visual encoder for semantic understanding with a spatial encoder for 3D structure reasoning. Extensive experiments show state-of-the-art performance across various visual spatial understanding and reasoning tasks.

Table 10: **Evaluation results on VSI-Bench.** For each metric, **bold** numbers indicate the best performance, while underlined numbers represent the second-best performance.

| Model | Numerical Question | | | | Multiple-choice Question | | | | Avg. |
|---|---|---|---|---|---|---|---|---|---|
| | Obj. Cnt | Abs. Dist. | Obj. Size | Room Size | Rel. Dist. | Rel. Dir. | Route Plan. | Appr. Order | |
| *Proprietary Models* | | | | | | | | | |
| GPT-4o | 46.2 | 5.3 | 43.8 | 38.2 | 37.0 | 41.3 | 31.5 | 28.5 | 34.0 |
| Gemini-2.0-Flash | 56.2 | 30.9 | **66.7** | 31.8 | **51.3** | 46.3 | 24.5 | **55.1** | 45.4 |
| *Open-Source Models* | | | | | | | | | |
| InternVL-2.5-4B | 45.0 | 15.5 | 37.5 | 24.6 | 37.2 | 41.5 | 31.4 | 26.2 | 32.6 |
| InternVL-2.5-8B | 50.6 | 31.3 | 40.2 | 39.3 | 45.1 | 41.4 | 29.4 | 43.9 | 40.2 |
| Kimi-VL-A3B | 41.3 | 30.4 | 42.1 | 13.2 | 26.3 | 32.6 | 32.0 | 11.2 | 28.7 |
| LLaVA-OneVision-7B | 46.1 | 26.2 | 36.3 | 29.5 | 30.8 | 37.2 | 35.1 | 21.8 | 33.1 |
| *Qwen2.5-VL-7B Based Spatial Models* | | | | | | | | | |
| Qwen2.5-VL-7B | 43.5 | 15.1 | 48.5 | 41.1 | 36.3 | 40.1 | 28.4 | 33.7 | 35.8 |
| SpaceR-7B | 63.2 | 30.0 | 60.3 | 37.6 | 39.7 | 45.6 | 31.4 | 48.2 | 44.5 |
| VILASR-7B | 63.5 | 34.4 | 60.6 | 30.9 | 48.9 | 45.2 | 30.4 | 49.2 | 45.5 |
| Video-R1 | 34.0 | 23.0 | 41.6 | 36.7 | 36.8 | 34.7 | 31.4 | 28.8 | 33.4 |
| *Qwen2.5-VL-3B Based Spatial Models* | | | | | | | | | |
| Qwen2.5-VL-3B | 32.9 | 22.1 | 17.3 | 31.5 | 32.8 | 44.2 | 26.3 | 28.5 | 29.4 |
| Spatial-MLLM-4B | **65.6** | **35.5** | 64.2 | 40.6 | 41.3 | **47.9** | 34.0 | 49.2 | **47.3** |
| SpatialLadder-3B | 63.5 | 34.3 | 61.7 | **43.9** | 45.4 | 44.8 | **35.6** | 36.4 | 45.7 |
| *Improvement* | *+30.6* | *+12.2* | *+44.4* | *+12.4* | *+12.6* | *+0.6* | *+9.3* | *+7.9* | *+16.3* |

Table 11: **Evaluation results on SPBench-SI.** For each metric, **bold** numbers indicate the best performance, while underlined numbers represent the second-best performance.

| Model | Numerical Question | | Multiple-choice Question | | Avg. |
|---|---|---|---|---|---|
| | Abs. Dist. | Obj. Size | Rel. Dist. | Rel. Dir. | |
| *Proprietary Models* | | | | | |
| GPT-4o | 19.7 | 29 | 81.3 | 39.2 | 42.4 |
| Gemini-2.0-Flash | 33.1 | 64.9 | 81.3 | 39.5 | 54.7 |
| *Open-Source Models* | | | | | |
| InternVL-2.5-4B | 27.3 | 36.2 | 73.6 | 33.0 | 42.5 |
| InternVL-2.5-8B | 15.6 | 40.8 | 76.9 | 35.6 | 42.3 |
| Kimi-VL-A3B | 11.3 | 40.2 | 62.6 | 27.1 | 35.3 |
| LLaVA-OneVision-7B | 23.6 | 27.2 | 54.9 | 27.1 | 33.2 |
| *Qwen2.5-VL-7B Based Spatial Models* | | | | | |
| Qwen2.5-VL-7B | 27.7 | 45.0 | **83.5** | 37.6 | 48.4 |
| SpaceR-7B | 8.4 | 62.9 | 80.2 | 42.8 | 48.6 |
| VILASR-7B | 10.3 | 63.0 | 81.3 | 46.1 | 50.2 |
| Video-R1 | 5.1 | 50.3 | 82.4 | 41.5 | 44.8 |
| *Qwen2.5-VL-3B Based Spatial Models* | | | | | |
| Qwen2.5-VL-3B | 30.9 | 17.8 | 75.8 | 36.6 | 40.3 |
| Spatial-MLLM-4B | 16.4 | 59.7 | 69.2 | 29.4 | 43.7 |
| SpatialLadder-3B | **45.5** | **71.7** | 81.3 | **82.4** | **70.2** |
| *Improvement* | *+14.6* | *+53.9* | *+5.5* | *+45.8* | *+29.9* |

Table 12: **Evaluation results on SPBench-MV.** For each metric, **bold** numbers indicate the best performance, while underlined numbers represent the second-best performance.

| Model | Numerical Question | | | Multiple-choice Question | | Avg. |
|---|---|---|---|---|---|---|
| | Obj. Cnt | Abs. Dist. | Obj. Size | Rel. Dist. | Rel. Dir. | |
| *Proprietary Models* | | | | | | |
| GPT-4o | 66.3 | 12.0 | 43.8 | 82.4 | 36.4 | 48.2 |
| Gemini-2.0-Flash | 49.9 | **40.7** | 65.1 | 76.5 | 25.0 | 51.4 |
| *Open-Source Models* | | | | | | |
| InternVL-2.5-4B | 65.1 | 24.0 | 23.9 | 82.4 | 20.5 | 43.2 |
| InternVL-2.5-8B | 50.0 | 25.0 | 37.0 | 88.2 | 6.8 | 41.4 |
| Kimi-VL-A3B | 13.7 | 23.3 | 33.0 | 76.5 | 38.6 | 37.0 |
| LLaVA-OneVision-7B | 21.1 | 21.3 | 19.2 | 76.5 | 22.7 | 32.2 |
| *Qwen2.5-VL-7B Based Spatial Models* | | | | | | |
| Qwen2.5-VL-7B | 46.1 | 11.0 | 35.4 | 88.2 | 11.4 | 37.3 |
| SpaceR-7B | 90.1 | 33.7 | 65.1 | 82.4 | 25.0 | 59.4 |
| VILASR-7B | 65.3 | 34.7 | 68.7 | 88.2 | 29.5 | 61.8 |
| Video-R1 | 33.9 | 18.0 | 45.6 | 76.5 | 29.5 | 40.7 |
| *Qwen2.5-VL-3B Based Spatial Models* | | | | | | |
| Qwen2.5-VL-3B | 36.7 | 14.7 | 14.9 | 88.2 | 18.2 | 36.6 |
| Spatial-MLLM-4B | 88.9 | 31.0 | 71.2 | 88.2 | 29.5 | 61.8 |
| SpatialLadder-3B | **94.9** | 34.7 | **76.4** | **100** | **50.0** | **71.2** |
| *Improvement* | *+58.2* | *+20.0* | *+61.5* | *+11.8* | *+31.8* | *+34.6* |

## C.3 Details of In-domain Benchmarks Results

We present the detailed evaluation results of VSI-Bench in Table 10. Our proposed SpatialLadder achieves an overall accuracy of 45.7%, surpassing all compared models except Spatial-MLLM, including those with 2–3 times larger parameter sizes. On average, SpatialLadder improves performance by 16.3% and demonstrates consistent gains across all sub-tasks of VSI-Bench. Notably, while Spatial-MLLM leverages an additional 3D encoder, our SpatialLadder relies solely on the vision encoder of Qwen2.5-VL-3B.

Furthermore, we report the detailed results of SPBench-SI and SPBench-MV in Tables 11 and 12, respectively. SpatialLadder attains 70.2% and 71.2% accuracy on these two benchmarks, corresponding to relative improvements of 29.9% and 34.6% over the base model Qwen2.5-VL-3B, and consistently outperforms all compared baselines.

Table 13: **Evaluation results on CV-Bench.** For each metric, **bold** numbers indicate the best performance, while underlined numbers represent the second-best performance.

| Model | 2D | | | 3D | Overall |
|---|---|---|---|---|---|
| | ADE20K | COCO | Avg. | Omni3D | |
| GPT-4o | 65.1 | 73.8 | 69.4 | 81.3 | 75.4 |
| InternVL-2.5-4B | 68.6 | 78.5 | 73.5 | 75.1 | 74.4 |
| Kimi-VL-A3B | 41.9 | 42.4 | 41.9 | 54.7 | 48.3 |
| LLaVA-OneVision-7B | 50.6 | 55.8 | 53.2 | 63.5 | 58.3 |
| Qwen2.5-VL-7B | **69.5** | **80.5** | **75.0** | **83.1** | **79.0** |
| Qwen2.5-VL-3B | 63.2 | 75.0 | 69.1 | 72.2 | 70.6 |
| SpatialLadder-3B | 67.1 | 77.6 | 72.4 | 74.9 | 73.7 |
| *Improvement* | *+3.9* | *+2.6* | *+3.3* | *+2.7* | *+3.1* |

Table 14: **Evaluation results on ViewSpatial-Bench.** For each metric, **bold** numbers indicate the best performance, while underlined numbers represent the second-best performance.

| Model | Camera Perspective | | | Person Perspective | | | | Overall |
|---|---|---|---|---|---|---|---|---|
| | Rel. Dir. | Obj. Ori. | Avg. | Obj. Ori. | Rel. Dir. | Sce. Sim. | Avg. | |
| GPT-4o | 41.5 | 19.6 | 33.7 | 41.2 | 32.8 | 21.9 | 31.5 | 32.6 |
| InternVL-2.5-4B | 37.1 | 31.8 | 40.8 | 43.6 | 37.1 | 26.1 | 35.1 | 37.9 |
| Kimi-VL-A3B | 26.9 | 22.1 | 25.1 | 63.1 | **43.9** | 20.3 | 41.5 | 33.6 |
| LLaVA-OneVision-7B | 29.8 | 26.1 | 28.5 | 22.4 | 31.0 | 26.9 | 26.5 | 27.5 |
| Qwen2.5-VL-7B | 47.8 | 30.9 | **41.8** | 41.6 | 35.4 | 26.9 | 39.8 | 37.9 |
| Qwen2.5-VL-3B | 43.5 | **32.5** | 39.5 | 40.0 | 29.9 | 26.3 | 32.0 | 35.6 |
| SpatialLadder-3B | **48.3** | 24.1 | 39.6 | **71.1** | 34.4 | **38.9** | **48.5** | **44.2** |
| *Improvement* | *+4.8* | *-8.4* | *+0.1* | *+31.1* | *+4.5* | *+12.6* | *+16.5* | *+8.6* |

## C.4 Details of Out-of-domain Benchmarks Results

We present the detailed evaluation results on CV-Bench in Table 13. Our proposed SpatialLadder achieves an overall performance of 73.7%, 5.3% below the best-performing baseline model. Nevertheless, it surpasses the base model Qwen2.5-VL-3B by 3.1%, demonstrating the robustness of our training framework in enhancing model performance. The detailed results on ViewSpatial-Bench are provided in Table 14, where SpatialLadder achieves an overall accuracy of 44.2%, outperforming all compared models and surpassing the base model by 8.6%. The detailed evaluation results on the out-of-domain benchmark SPAR-Bench are provided in Table 2 in the main text.

## C.5 SFT Training Stability

We show in Figure 9 the training dynamics of Stage 1–2 SFT, including loss and mean token accuracy. The results indicate that the model trains smoothly and converges rapidly in both stages, demonstrating the high quality and consistency of SpatialLadder-26*k* and ensuring stable performance during full-model fine-tuning.

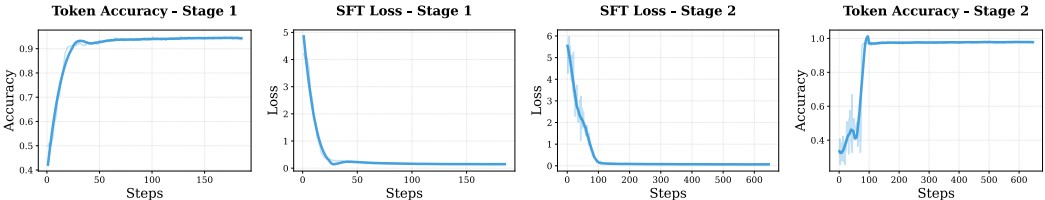

Figure 9: Training dynamics of loss and mean token accuracy during stage 1-2

## C.6 DETAILS OF SEMANTIC ENTROPY.

To quantify response diversity for uncertainty analysis, we introduce semantic entropy as a clustering-based metric. For each question $q$, we sample responses $\{o_1, o_2, ..., o_G\}$ at temperature 0.9 (8 samples per question) and partition them into semantic clusters $\mathcal{C} = \{C_1, C_2, ..., C_K\}$ based on accuracy rewards, where $C_k = \{o_i : \mathcal{R}(o_i) = r_k\}$ for distinct reward values $\mathcal{R} = \{r_1, r_2, ..., r_K\}$. Semantic entropy is then computed as:

$$\text{SE}(q) = -\sum_{i=1}^{K} p_i \log p_i, \quad \text{where } p_i = \frac{|C_i|}{N} \tag{9}$$

This measure captures the distributional diversity of semantically distinct response clusters, providing a principled approach to quantify model uncertainty beyond surface-level textual variations.

## C.7 DETAILS OF ATTENTION ANALYSIS.

For visual attention analysis, we use two metrics, Visual Attention IoU, used to measures attention concentration within object bounding boxes, defined as:

$$\text{IoU}_{\text{att}} = \frac{\sum_{i \in B_{\text{obj}}} \hat{a}_i}{\sum_{j=1}^{N} \hat{a}_j} \tag{10}$$

where $B_{\text{obj}}$ represents visual tokens within the target object's bounding box and $\hat{a}_i$ denotes min-max normalized attention weights. Visual Attention Entropy measures attention concentration:

$$H_{\text{att}} = -\sum_{i=1}^{N} p_i \log p_i \tag{11}$$

where $p_i$ represents the probability distribution from normalized attention weights.

In Figure 13, we present additional comparisons of attention distributions between SpatialLadder and its base model Qwen2.5-VL-3B on the object size estimation task. The results demonstrate that, compared with the base model, SpatialLadder exhibits a more focused allocation of attention on task-relevant objects. This indicates that our training framework effectively guides the internal attention distribution of the model, thereby enhancing its inherent perceptual ability and supporting more reliable performance in spatial reasoning tasks.

## D ADDITIONAL EXPERIMENTS

### D.1 CHAIN-OF-THOUGHT TRAINING DYNAMICS

The analysis of chain-of-thought reasoning dynamics, as illustrated in Figure 10, provides additional insights into the importance of explicit reasoning processes in spatial understanding tasks. While both the full model with chain-of-thought and its variant without reasoning components achieve comparable performance in terms of accuracy reward curves during later training stages, significant differences emerge in training stability and convergence patterns.The chain-of-thought enabled model demonstrates superior training stability, characterized by faster reduction in reward standard deviation and smoother convergence behavior. More critically, the actual performance trajectories

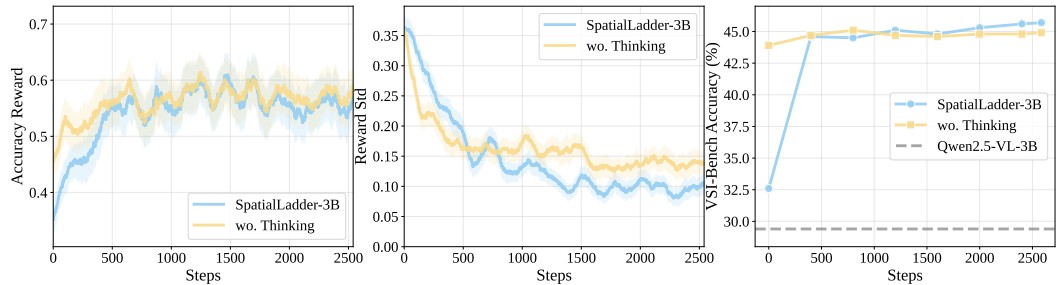

Figure 10: **Impact of thinking during training.** Left: accuracy rewards; Middle: reward standard deviation; Right: VSI-Bench performance comparison.

Table 15: **Performance comparison across spatial reasoning datasets.** SI, MV, and VID denote single-image, multi-view, and video modalities, respectively.

| Dataset | Modality | Size | VSI-Bench |
|---|---|---|---|
| SpaceR-151$k$ (Ouyang et al., 2025) | VID | 151,310 | 35.1 |
| Spatial-MLLM-120$k$(Ouyang et al., 2025) | MV+VID | ≈120,000 | 40.0 |
| SpatialLadder-26$k$ | **SI+MV+VID** | **26,610** | **43.9** |

reveal distinct learning dynamics: the model without chain-of-thought reasoning reaches an early performance plateau and exhibits limited improvement thereafter, whereas the chain-of-thought variant maintains continuous performance enhancement throughout training, ultimately achieving superior final accuracy on evaluation benchmarks.

## D.2 COMPARISON WITH OTHER SPATIAL DATASET

Table 15 demonstrates the effectiveness of our dataset design through comparative analysis with existing spatial reasoning datasets. All models are trained using supervised fine-tuning on Qwen2.5-VL-3B as the base model to ensure fair comparison. Despite utilizing significantly fewer training samples (26,610 vs. 151,310 and ≈120,000), SpatialLadder-26$k$ achieves superior performance on VSI-Bench, reaching 43.9% accuracy compared to 35.1% for SpaceR-151k and 40.0% for Spatial-MLLM-120k. This performance gain is attributed to our comprehensive approach that integrates object localization tasks and spatial reasoning tasks across single-image, multi-view, and video modalities within a unified framework, contrasting with previous datasets that focus on individual modalities or limited combinations. The results validate that strategic dataset curation and progressive training can achieve better spatial reasoning capabilities with substantially reduced data requirements, highlighting the importance of data quality and training methodology over sheer dataset scale.

## D.3 DATASET SCALING ANALYSIS

Figure 11 demonstrates the consistent scaling potential of our dataset across spatial reasoning benchmarks using Qwen2.5-VL-3B as the base model for supervised fine-tuning. Overall performance increases steadily from 36.2% to 60.2%, while VSI-Bench improves from 29.4% to 43.9% as dataset scaling progresses from 0% to 100%. The sustained upward trajectories without saturation at full scale indicate substantial room for further improvement through continued dataset expansion. These scaling patterns validate the effectiveness of our dataset design at larger scales and highlight the potential for achieving even stronger spatial reasoning capabilities through strategic dataset augmentation.

## D.4 IMPACT OF PROGRESSIVE TRAINING SEQUENCE

To validate our training paradigm, we conduct ablation studies using Qwen2.5-VL-3B as the base model with supervised fine-tuning protocols. Figure 12 highlights the critical importance of training order in developing spatial reasoning capabilities. Our progressive perception-to-spatial training paradigm achieves 43.9% accuracy on VSI-Bench, outperforming both spatial-only training (42.7%)

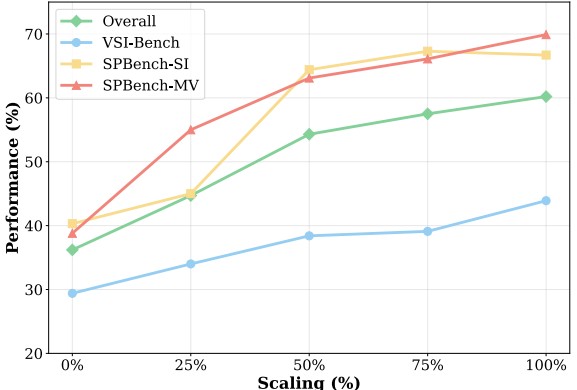
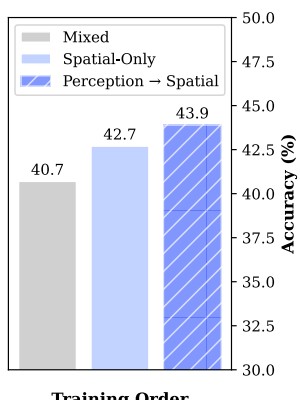

Figure 11: Dataset scaling analysis across spatial reasoning benchmarks.

Figure 12: VSI-Bench accuracy across different training order.

and mixed training approaches (40.7%). Crucially, the mixed training performance is even lower than the stage 2-only baseline, indicating that naively combining simple object localization tasks (Stage 1) with complex reasoning tasks (Stage 2) results in task interference.

The sequential approach that first establishing perceptual foundations through localization tasks, followed by spatial reasoning training, yields a 1.2% improvement over direct spatial training and a 3.2% gain over simultaneous mixed training. These results validate our hypothesis that systematic progression from basic perception to complex reasoning creates more robust spatial understanding than alternative strategies. Our progressive structure effectively decouples conflicting objectives, establishing a stable perceptual foundation first to facilitate the learning of complex spatial reasoning, and underscores that structured skill development through sequential training stages is essential for optimal spatial reasoning capabilities.

Table 16: **VSI-Bench performance comparison across different KL weight and reward scaling.** Here, $\beta_{\mathrm{KL}}$ denotes the KL weight, while $w_{\mathrm{format}}$ and $w_{\mathrm{accuracy}}$ represent the scaling factors for the format and accuracy rewards, respectively.

| $\beta_{\mathbf{KL}}$ | $w_{\mathbf{format}}$ | $w_{\mathbf{accuracy}}$ | **VSI-Bench** |
|------|------|------|------|
| 0.01 | 1.0 | 1.0 | **45.7** |
| 0.01 | 0.8 | 0.2 | 44.9 |
| 0.01 | 1.0 | 1.0 | 45.2 |
| 0.04 | 1.0 | 1.0 | 45.2 |

## D.5 RL CONFIGURATION ANALYSIS

In our three-stage training framework, the reward coefficients and RL configurations are closely aligned with previously validated robust settings (Guo et al., 2025; Shen et al., 2025; Huang et al., 2025). This design isolates the impact of our training data and framework from the RL hyperparameters, allowing us to demonstrate the intrinsic effectiveness of our method. To ensure completeness and reproducibility, we conducted additional experiments to analyze the sensitivity of the RL configuration, specifically the KL weight and reward scaling (Table 16). While our default configuration yields optimal results, the model demonstrates strong robustness to variations in hyperparameters, maintaining consistently high performance across different settings.

Empirically, we observe that the format reward converges rapidly during the early training stages. Under this premise, the robustness to reward weights is an inherent property of the GRPO mechanism: its group-based advantage normalization effectively cancels out the absolute scaling of the accuracy reward once the format reward stabilizes. This provides a theoretical explanation for why our RL stage is robust to variations in reward scaling.

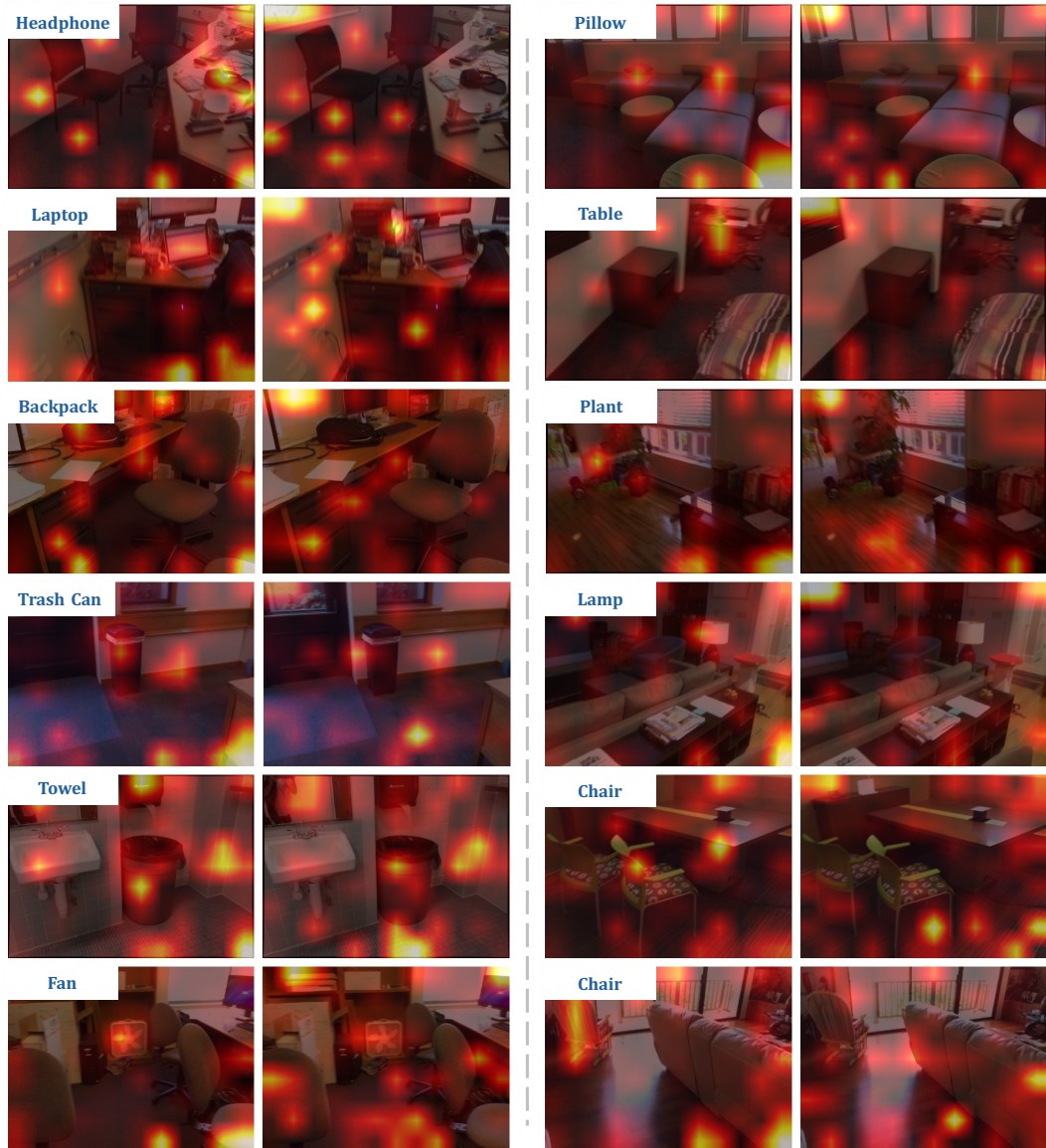

Figure 13: **Additional examples of attention distribution comparison.** For each example, the left panel shows the attention distribution of SpatialLadder, while the right panel shows that of Qwen2.5-VL-3B.

## E  LIMITATIONS AND FUTURE WORK

Our work presents several limitations. Due to computational resource constraints, our experiments are conducted exclusively on 3B-parameter models, leaving the scalability to larger models unexplored. Additionally, our SpatialLadder-26$k$ dataset has substantial room for scaling, with only 26,610 samples that may be insufficient for capturing the full complexity of spatial reasoning scenarios. The dataset's reliance primarily on ScanNet scenes also introduces domain bias toward indoor environments, limiting generalization to diverse real-world scenarios. Furthermore, our three-stage progressive training framework follows a fixed sequential structure that may not be optimal for all spatial reasoning tasks, lacking the flexibility to adapt to task-specific requirements.

These limitations suggest promising directions for future work. Scaling our progressive training approach to larger models (7B, 13B, and beyond) could reveal additional performance gains and better understand the scalability of hierarchical spatial learning. Expanding the dataset both in

scale and diversity—incorporating larger sample sizes, outdoor landscapes, urban environments, and domain-specific imagery—would likely yield better performance and enhance robustness across varied scenarios. Additionally, developing adaptive training frameworks that dynamically adjust learning sequences based on task characteristic or model performance could improve efficiency. Finally, validating SpatialLadder in real-world applications such as robotics navigation and autonomous driving would provide valuable insights into practical deployment and identify areas for further improvement.

## LLM USAGE

In this section, we clarify the role of large language models (LLMs) in preparing this work. We acknowledge that LLMs were employed exclusively for writing assistance and linguistic refinement in the preparation of this manuscript. These tools were utilized to enhance the clarity, grammatical accuracy, and academic style of the text while preserving all original research contributions, methodological approaches, and scientific insights developed by the authors. The language models served solely as writing aids to improve sentence structure, enhance readability of technical content, refine academic terminology, and ensure consistency in writing style throughout the manuscript.

It is important to emphasize that LLMs were not employed for research ideation, conceptual development, literature review, citation discovery, data analysis, experimental design, or generation of research hypotheses and conclusions. All research ideas, experimental work, data analysis, and scientific conclusions presented in this paper originate entirely from the authors' independent intellectual work. The use of LLMs was limited to linguistic enhancement and does not constitute contribution at the level of authorship. The authors take full responsibility for all content, including any text that was refined with LLM assistance, ensuring that the core intellectual contributions and scientific merit of this work remain wholly attributable to the listed authors.

