# OpenReview forum: "SpatialLadder: Progressive Training for Spatial Reasoning in Vision-Language Models"
_ICLR.cc/2026/Conference — ICLR 2026 Poster_

### Official Review · Reviewer_g2My · 2025-10-30

**Soundness:** 4
**Presentation:** 3
**Contribution:** 2
**Rating:** 6
**Confidence:** 4

**Summary:**

This paper introduces SpatialLadder, a new 3B VLM in 3D spatial reasoning with a 3-stage training scripts, data collection pipeline and strong task performance. By incorporating this 3-stage training, SpatialLadder learns spatial knowledge and reasoning progressively and tops at spatial benchmarks while performing decently on OOD spatial tasks.

**Strengths:**

1. The 3-stage training pipeline looks reasonable and effective with carefully curated 26K data spanning across 4 domains. The data would be potentially beneficial to the community
2. SpatialLadder-3B tops at in-domain and OOD spatial benchmarks
3. Comprehensive ablations and visual analysis about attention maps and case study

**Weaknesses:**

1. Lack of ablation on the 3-stage training paradigm. As both stage 1 and 2 are SFT-ing over spatial data, though the type of questions and visual input maybe different, but what if combining the first two SFT stages into one as they both serve as broadening the spatial knowledge of base VLMs
2. Only one model was trained. Will the model perform better if using another VLM or increasing the parameter size to 7B or larger?

**Questions:**

I’m curious about how VLM performs on general VL benchmarks that are partially related to spatial reasoning like RealWorldQA, RoboSpatial-Home, MME-RealWorld-Lite. Will this 3-stage introduce significant catastrophic forgetting to VLM that hinders the general performance of the model?

---

> ### Author Response · Authors · 2025-11-20
> **Rebuttal - Part I**
>
> We sincerely thank Reviewer g2My for the positive assessment and constructive feedback. We appreciate the recognition of the effectiveness of our 3-stage training pipeline and the value of the SpatialLadder-26k dataset to the community. We have carefully addressed the reviewer's questions and concerns as follows.
>
> ### W1: Lack of ablation on the 3-stage training paradigm.
>
> We appreciate the reviewer for emphasizing this critical comparison and fully agree that comparing against a joint training baseline is essential. We respectfully clarify that this exact comparison was conducted and reported in **Appendix D.4 (Fig. 13)**, where we performed an ablation study using identical data to strictly compare progressive training against joint (mixed) training. We highlight two critical findings from this experiment:
>
> - **Progressive Training Significantly Outperforms Joint Training:** The results strongly support our claim: our progressive training strategy achieves the highest performance on VSI-Bench (**43.9%**), outperforming the joint training baseline (“Mixed”, **40.7%**). This demonstrates that simply merging different tasks into a single stage or performing joint fine-tuning (**40.7%**) yields substantially worse results compared to our progressive pipeline (**43.9%**). These findings highlight the **critical importance of training tasks in a stage-wise, progressively structured manner**.
>
> - **Evidence of Task Interference:** Crucially, the mixed training performance (**40.7%**) is even lower than the stage 2 only baseline (**42.7%**) . This indicates that naively combining simple object localization tasks (Stage 1) with complex reasoning tasks (Stage 2) leads to **task interference** . Our progressive structure is therefore necessary: it effectively decouples these conflicting objectives, **establishing a stable perceptual foundation first to facilitate the learning of complex spatial reasoning**.
>
> ### W2: Only one model was trained. Will the model perform better if using another VLM or increasing the parameter size to 7B or larger?
>
> We sincerely appreciate the reviewer's concern regarding the generalizability and scalability of our framework. To empirically validate that our  progressive training principle is effective beyond a single model configuration, we conducted new ablation experiments on three additional models: **InternVL-2.5-2B** [1] and **LLaVA-Next-Video-7B** [2] (to test architectural generalizability), and **Qwen2.5-VL-7B** (to test scalability to larger parameters). We reproduced the ablation in **Appendix D.4 (Fig. 13)** study comparing our sequential training strategy against the joint training strategy in Stage 1-2 SFT. We have updated our paper to include the experiments in **Fig. 9**, highlighted in blue:
>
> | Model Architecture      | Base Model | + Joint Training | + Sequential Training | Improvement (Seq. vs. Joint) |
> | ----------------------- | ---------- | ---------------- | --------------------- | ---------------------------- |
> | **InternVL-2.5-2B**     | 23.4       | 38.9             | **41.7**              | **+2.8**                     |
> | **LLaVA-Next-Video-7B** | 34.9       | 42.9             | **44.5**              | **+1.6**                     |
> | **Qwen2.5-VL-7B**       | 35.8       | 42.1             | **44.8**              | **+2.7**                     |
>
> - **Universal Effectiveness across Diverse Architectures** The results provide compelling evidence for the universality of our framework. Across diverse architectures (InternVL, LLaVA, and Qwen), our sequential training strategy consistently outperforms the joint training baseline by a stable margin (**+1.6%** to **+2.8%**). This consistency confirms that the "Perception-Reasoning Gap" is an intrinsic bottleneck in spatial learning that exists independently of the specific model architecture. **Our progressive solution effectively addresses this universal challenge, yielding improvements without requiring architecture-specific tuning**.
> - **Scalability to Larger Foundation Models:** Regarding scalability, the results on Qwen2.5-VL-7B demonstrate that our method remains highly effective as model size increases. With only SFT data, the 7B model achieves a remarkable **44.8%** accuracy under our sequential framework, significantly outperforming the joint training baseline (**42.1%**). This indicates that **our progressive training principle scales effectively, unlocking greater potential in stronger foundation models** and suggesting that the benefits of structured learning do not diminish with increased parameter scale.

---

> ### Author Response · Authors · 2025-11-20
> **Rebuttal - Part II**
>
> ### Q1: Will this 3-stage introduce significant catastrophic forgetting to VLM that hinders the general performance of the model?
>
> We thank the reviewer for this crucial question regarding the potential risk of "catastrophic forgetting". We fully agree that preserving general VLM capabilities is essential. To strictly address this, we conducted a comprehensive evaluation on the exact benchmarks suggested by the reviewer (**RealWorldQA**, **RoboSpatial-Home**, **MME-RealWorld-lite**), alongside standard general benchmarks (**MMMU** [3], **MMBench** [4]) to ensure a rigorous assessment of broad capabilities. The results are summarized below:
>
> | Method           | RealWorldQA | RoboSpatial-Home | MME-RealWorld-lite | MMMU | MMBench | Avg.     |
> | ---------------- | ----------- | ---------------- | ------------------ | ---- | ------- | -------- |
> | Qwen2.5-VL-3B    | 58.8        | 14.9             | 38.6               | 48.3 | 83.3    | 48.8     |
> | SpatialLadder-3B | 57.5        | **47.4**         | 37.9               | 47.1 | 82.4    | **54.5** |
> | Delta            | -1.3        | **+32.5**        | -0.7               | -1.2 | -0.9    | **+5.7** |
>
> The experimental results indicate that:
>
> - **No Catastrophic Forgetting on General Tasks:** The experimental results indicate stable performance across general domains. On broad benchmarks like MMBench, MMMU, RealWorldQA, and MME-RealWorld-Lite, we observe only negligible performance fluctuations (ranging from **-0.7% to -1.3%**). This indicates that **the model’s core multimodal capabilities are well-preserved**. This minor trade-off is typical for domain-specialized fine-tuning and clearly demonstrates that our 3-stage training does not constitute catastrophic forgetting.
> - **Exceptional Generalization to RoboSpatial-Home:** We are particularly grateful to the reviewer for suggesting RoboSpatial-Home. Contrary to the concern of performance degradation, our model achieved a **massive improvement of +32.5%** on this benchmark (**14.9% $\to$ 47.4%**). This gain is strictly driven by our model's enhanced understanding of fine-grained spatial relationships, which are critical for robotics tasks. Specifically, we observed significant gains in the sub-tasks of **Spatial Compatibility (17.1% $\to$ 78.1%)** and **Spatial Configuration (27.6% $\to$ 67.5%)**. This result serves as powerful evidence that our spatial training not only "remembers" general knowledge but **significantly empowers the model to handle complex spatial reasoning tasks**.
>
> ### Reference
>
> [1] Expanding Performance Boundaries of Open-Source Multimodal Models with Model, Data, and Test-Time Scaling.
>
> [2]LLaVA-NeXT-Interleave: Tackling Multi-image, Video, and 3D in Large Multimodal Models
>
> [3] MMMU: A Massive Multi-discipline Multimodal Understanding and Reasoning Benchmark for Expert AGI. CVPR 2024
>
> [4] MMBench: Is Your Multi-modal Model an All-around Player? ECCV 2024
>
> -----
>
> We sincerely thank Reviewer g2My once again for the thoughtful and constructive feedback, as well as for the recognition of our work. Your insights have motivated us to conduct additional experiments and refine the manuscript, which we believe has further strengthened the contributions of our study. We remain fully available to address any additional questions or suggestions.

---

> > ### Comment · Reviewer_g2My · 2025-11-24
> >
> > Thank you for detailed responses, which have addressed most of my concerns. I would thus keep my score of acceptance.

---

> ### Author Response · Authors · 2025-11-25
>
> Dear reviewer g2My,
>
> Thank you very much for the positive feedback and continued support. We are glad that our responses have addressed your concerns. We deeply appreciate your constructive engagement with our work and welcome any further questions or suggestions you may have.

---

### Official Review · Reviewer_eN7r · 2025-10-31

**Soundness:** 4
**Presentation:** 4
**Contribution:** 3
**Rating:** 8
**Confidence:** 4

**Summary:**

This paper tackles the challenge of spatial reasoning in Vision-Language Models (VLMs), positing that current models fail because they attempt to learn this skill monolithically rather than hierarchically. The authors introduce a "perception-reasoning gap" and propose a systematic, progressive approach to bridge it. Their contributions are twofold: 1) A new multimodal dataset, SpatialLadder-26k, containing over 26,000 samples for object localization, single-image, multi-view, and video spatial reasoning, built with a standardized pipeline. 2) A three-stage progressive training framework designed to first establish perceptual grounding (object localization via supervised fine-tuning), then develop spatial understanding across multiple dimensions and modalities, and finally strengthen complex reasoning using reinforcement learning (GRPO) with chain-of-thought. The resulting 3B parameter model, SpatialLadder, achieves state-of-the-art performance, significantly outperforming its base model as well as larger proprietary models like GPT-4o on a suite of in-domain and out-of-domain benchmarks. Extensive ablation studies validate that the progressive, curriculum-based approach is crucial to the model's success.

**Strengths:**

1. **Clear and Well-Motivated Hypothesis:** The paper is built around the intuitive and compelling idea that spatial intelligence is hierarchical and must be learned progressively from perception to reasoning. This "perception-reasoning gap" hypothesis is clearly articulated and effectively motivated by a preliminary experiment showing that adding perceptual cues improves a base model's performance.
2. **High-Quality Methodology:** The work introduces a comprehensive solution with two strong components. The SpatialLadder-26k dataset is systematically designed to cover a full spectrum of spatial tasks, addressing the fragmentation of existing resources. The three-stage training framework is a logical and direct implementation of the paper's core hypothesis, providing a clear recipe for building spatial reasoning capabilities.
3. **Extensive and Rigorous Evaluation:** The experimental validation is exceptionally thorough. The authors evaluate their model on six different benchmarks, assessing both in-domain and, crucially, out-of-domain generalization. The comparison against a strong suite of open-source and proprietary models clearly demonstrates the effectiveness of their approach.
4. **Convincing Ablation Studies:** The paper includes a comprehensive set of ablation studies that systematically dismantle the proposed framework to prove the value of each component. The analysis confirms the importance of each training stage, the multimodal data, the chain-of-thought mechanism, and, most importantly, the progressive training order itself. The dataset comparison in the appendix, which shows their smaller, curated dataset outperforms larger ones, is a particularly strong piece of evidence for their approach.

**Weaknesses:**

1. **Limited Model Scale Exploration:** The experiments are confined to a 3B parameter model. While the results are impressive for this model size, the paper does not explore whether these significant gains will translate to larger, state-of-the-art models (e.g., 7B+).
2. **Dataset Domain Bias:** The dataset's reliance on the indoor scenes from ScanNet introduces a potential domain bias. The model's excellent performance might not fully generalize to outdoor environments or other visually distinct domains.
3. **Incremental Novelty of Components:** While the overall system and the curriculum are novel and highly effective, the individual techniques used (supervised fine-tuning, GRPO for reinforcement learning) are established methods. The main contribution lies in the successful and systematic application of these components in a progressive manner to solve the spatial reasoning problem, rather than the invention of a new technique.

**Questions:**

1. The ablation study (Figure 5) indicates that Stage 3 (Reinforcement Learning) contributes less to the final performance (2.1% drop upon removal) compared to Stage 2 (Understanding SFT, 9.4% drop). Could you elaborate on the specific role of the RL stage? Does it primarily refine the structure and logical flow of the reasoning chains, or does it enable the model to solve problems that were previously intractable after Stage 2?
2. The comparison in Appendix D.2 showing that the smaller SpatialLadder-26k dataset is more effective than larger datasets is a very strong result. Based on your analysis, which aspect of your dataset design do you believe is the most critical for this data efficiency: the inclusion of foundational localization tasks, the multi-modal coverage (SI+MV+Video), or the structured nature of the QA pairs?
3. The proposed framework uses a fixed, sequential training order. Have you explored or considered more dynamic training curricula? For example, could there be benefits to cyclically reintroducing tasks from earlier stages or using a mixed-task approach once a foundational capability in each stage is achieved?
4. The visual attention analysis is quite insightful. Have you investigated whether the more focused attention in SpatialLadder translates to better performance on multi-object relational tasks? For instance, when asked for the relative direction between two objects, does the model's attention map show precise focus on both entities simultaneously?

---

> ### Author Response · Authors · 2025-11-20
> **Rebuttal - Part I**
>
> We sincerely thank Reviewer eN7r for the encouraging assessment and the recommendation of acceptance. We are particularly grateful for the recognition of our core insight, systematic methodology and rigorous experiments . We have carefully addressed the reviewer's questions and concerns as follows.
>
> ### W1: Limited Model Scale Exploration.
>
> We sincerely appreciate the reviewer's suggestion to verify our findings on larger model scales. While we initially discussed this potential in Appendix E. To empirically verify whether our progressive framework scales to lager models, we conducted new ablation experiments on **Qwen2.5-VL-7B** during the rebuttal. We reproduced the key comparison from **Appendix D.4 (Fig. 13)**, comparing our sequential training strategy against the joint training baseline in Stage 1-2 SFT. We have updated our paper to include the experiments in **Fig. 9**, highlighted in blue:
>
> | Model Architecture | Base Model | + Joint Training | + Sequential Training | Improvement (Seq. vs. Joint) |
> | ------------------ | ---------- | ---------------- | --------------------- | ---------------------------- |
> | **Qwen-2.5-VL-7B** | 35.8       | 42.1             | **44.8**              | **+2.7**                     |
>
> The results indicate that:
>
> - **Significant Performance Gains on 7B Scale:** Our framework successfully scales to the larger model, boosting the Qwen2.5-VL-7B performance from **35.8%** to **44.8%** with only SFT. This substantial improvement confirms that the benefits of our approach are not confined to the 3B parameter regime but remain highly effective for stronger foundation models.
> - **Universal Superiority of Sequential Strategy** Crucially, the results reaffirm the necessity of our training topology even at this scale. Consistent with our 3B experiments, the sequential training strategy significantly outperforms the joint training baseline (**44.8% vs. 42.1%**). This **+2.7% margin** validates that the "Perception-Reasoning Gap" is an intrinsic bottleneck that scaling parameters alone does not solve, and that **our progressive strategy remains the optimal solution for unlocking the full potential of larger VLM architectures**.
>
> ### W2: Dataset Domain Bias.
>
> We thank the reviewer for raising this crucial concern. Ensuring that our model acquires generalized spatial intelligence rather than merely memorizing indoor layouts was a central consideration of our work.
>
> To address this, we clarify the strict data separation in our existing results and provide new experimental evidence on two challenging OOD benchmarks (**MMSI-Bench** [1] and **MindCube** [2])  (updated in **Tab. 2** of our paper) to demonstrate robustness across diverse domains.
>
> - **Evidence from Existing OOD Benchmarks:** First, we highlight that our evaluation in **Tab. 2** already addresses domain generalization. **CV-Bench** consists entirely of scenes from **ADE20K, COCO, and Omni3D**, covering a wide range of **outdoor environments** and diverse visual domains completely unrelated to ScanNet. ViewSpatial-Bench also incorporates a substantial number of images from MS-COCO. Our model achieves consistent gains on these benchmarks (e.g., **+3.1%** on CV-Bench, **+8.6%** on ViewSpatial-Bench), providing **initial evidence of generalization beyond indoor scanning data**.
>
> - **Generalization on MMSI-Bench and MindCube:** To definitively answer the concern about "general spatial intelligence," we evaluated SpatialLadder on two additional, highly challenging OOD benchmarks that differ significantly from our training data:
>
>   - **MMSI-Bench:** This benchmark aggregates data from 8 diverse sources, including **nuScenes, Waymo** (autonomous driving), and **Ego4D** (egocentric robotics). It forces the model to reason in entirely unseen **outdoor and dynamic environments**, with QA formats completely distinct from our templates.
>   - **MindCube:** This benchmark evaluates complex spatial cognition (e.g., mental rotation) using scenes from **DL3DV-10K, WildRGB-D, and ArkitScenes**, covering a wide range of **wild indoor/outdoor** RGB-D data. It tests deep spatial understanding beyond simple localization.
>
>   As shown in the table below, SpatialLadder-3B consistently outperforms the base model on these distinct domains:
>
>   | Method               | MMSI-Bench | MindCube |
>   | -------------------- | ---------- | -------- |
>   | **Qwen2.5-VL-3B**    | 26.5       | 37.6     |
>   | **SpatialLadder-3B** | **29.2**   | **43.4** |
>   | **Improvement**      | **+2.7**   | **+5.8** |
>
>   The consistent gains across diverse scenes (MMSI, **+2.7%**) and complex mental reasoning tasks (MindCube, **+5.8%**) strongly suggest that our progressive framework teaches the model **abstract spatial logic** that **generalizes effectively beyond the specific textures and layouts of indoor ScanNet scenes**.

---

> ### Author Response · Authors · 2025-11-20
> **Rebuttal - Part II**
>
> ### W3: Incremental Novelty of Components.
>
> We thank the reviewer for accurately characterizing the nature of our contribution. We fully agree that our contribution is not merely a combination of established techniques (SFT, RL), but rather the **identification of a critical bottleneck in current VLMs** and the design of a **systematic framework** specifically engineered to resolve it. While the optimization tools are standard, the architecture of the learning process is novel and essential.
>
> We respectfully posit that this "systematic application" constitutes a significant scientific contribution for two reasons:
>
> - **Quantifying the "Perception-Reasoning Gap":**  Unlike existing works that treat spatial reasoning as a monolithic capability, we provide a quantitative diagnosis identifying the "Perception-Reasoning Gap" as a primary bottleneck in current VLM spatial reasoning.  As shown in our preliminary analysis (**Fig. 1**), simply injecting perceptual hints boosts the base model's accuracy from **36.5%** to **46.0%** without any model updates. This is a key insight: models often possess the reasoning logic but **lack the perceptual grounding to activate spatial intelligence**.
> - **Structural Decomposition vs. Naive Combination:** Guided by this diagnosis, SpatialLadder is not a generic application of curriculum learning. Instead, it systematically decouples spatial intelligence into a functional hierarchy: **Perception (Stage 1) $\to$ Understanding (Stage 2) $\to$ Complex Reasoning (Stage 3)**. This differs from standard multimodal fine-tuning. We do not simply feed data from easy to hard; **we explicitly align the training stages with the inherent dependencies of spatial processing**, ensuring the model can "see" (localize) before it tries to "measure" (spatial understanding) or "deduce" (reasoning).
> - **Validation of the Framework:** Our ablation study in **Appendix D.4 (Fig. 13)** validates that this structural design is essential. Under a strict control setting with identical SFT data volume, our progressive framework achieves **43.9%** on VSI-Bench, consistently outperforming a naive mixed-data training strategy (**40.7%**). This confirms that our performance gains stem from the **structured formulation derived from our core insight, rather than merely from data aggregation or naive method combination**.

---

> ### Author Response · Authors · 2025-11-20
> **Rebuttal - Part III**
>
> ### Q1: Could you elaborate on the specific role of the RL stage?
>
> We thank the reviewer for this insightful question. While the absolute improvement of the final stage may appear modest (+2.1%), our experimental evidence demonstrates that this stage is essential for surpassing the inherent performance ceiling of SFT (Stage 2). Beyond accuracy gains, it fundamentally reshapes the model’s reasoning dynamics, enabling the emergence of more consistent, confident, and higher-quality reasoning chains:
>
> - **Surpassing the SFT Ceiling:** To rigorously determine whether the gains in Stage 3 stem from the RL algorithm itself or simply from extended SFT training steps, we conducted a controlled experiment comparing **Extended SFT** (adding an extra epoch to Stage 2) against our Stage 3 RL. We have updated our paper to include the experiments in **Tab. 3**, highlighted in blue:
>
>   | Model                               | VSI-Bench | SPBench-SI | SPBench-MV | Avg.                         |
>   | ----------------------------------- | --------- | ---------- | ---------- | ---------------------------- |
>   | Qwen2.5-VL-3B (Backbone)            | 29.4      | 40.3       | 36.6       | 38.8                         |
>   | + Stage 1-2  SFT Training           | 43.9      | 68.5       | 69.9       | 60.2                         |
>   | + Stage 1-2 (2 Epochs) SFT Training | 43.4      | 64.3       | 67.0       | 58.2 **($\downarrow$ 2.0%)** |
>   | + Stage 1-3 Training (Ours)         | **45.7**  | **70.2**   | **70.9**   | **62.3 ($\uparrow$ 2.1%)**   |
>
>   The comparison shows that additional SFT actually leads to a consistent drop in evaluation performance (**60.2% → 58.2%**), indicating that extra supervised training can cause overfitting and degrade results. In contrast, Stage 3 RL not only avoids performance degradation but achieves a substantial improvement (**60.2% → 62.3%**), demonstrating that reinforcement learning **plays a crucial role in the overall performance gains**, which is not achievable with merely SFT.
>
> - **Transforming Exploration into Conviction:** We quantify the fundamental shift in "Reasoning Mode" driven by RL using Semantic Entropy [3] (**Fig. 6**), a metric quantifying model uncertainty by measuring the diversity of semantically distinct response clusters. During the SFT phases (Stages 1-2), the entropy rises (**1.24 $\to$ 1.47**), indicating the model is expanding its reasoning space and learning diverse possibilities. However, Stage 3 RL drastically reduces this entropy (**$\to$ 0.66**), incentivizing the model to collapse the solution space into confident, consistent reasoning paths (Consolidation). This quantitative dynamic demonstrates that RL performs a unique function: it does not merely repeat the training data but actively **refines and stabilizes** the reasoning process, **transitioning the model from broad exploration to focused conviction**, which aligns with the phenomena observed in recent studies [4].
>
> - **Structured Reasoning and Self-Verification:** Regarding the specific reasoning abilities improved, the GRPO phase cultivates **structured reasoning and self-correction behaviors** that go beyond simple formatting or answer consistency. As visualized in **Fig. 8**, the model exhibits **systematic decomposition**, breaking complex spatial queries into logical steps rather than generating unstructured descriptions. Crucially, the model develops **self-verification patterns** that are effectively incentivized by the RL reward structure. These behaviors enhance the transparency and auditability spatial reasoning process that SFT alone fails to consistently establish.

---

> ### Author Response · Authors · 2025-11-20
> **Rebuttal - Part IV**
>
> ### Q2: Based on your analysis, which aspect of your dataset design do you believe is the most critical for this data efficiency?
>
> We thank the reviewer for highlighting this strong result. We agree that the superior efficiency of SpatialLadder-26k (outperforming datasets 5-6 times larger) is a critical finding. Based on our ablation studies and dataset construction analysis, we believe this high data efficiency stems from the synergy of three specific design choices:
>
> - **Explicit Integration of Foundational Perception Tasks:** The most distinct factor is the inclusion of pure object localization data. As demonstrated in our ablation study (**Appendix D.4, Fig. 13**), simply adding these foundational perception tasks enables the model to achieve **43.9%** accuracy, outperforming a model trained on spatial reasoning data alone (**42.7%**). This confirms that dedicating a portion of the data budget to enhance perception is more efficient than spending the entire budget on reasoning training, as **it establishes the necessary perceptual grounding for complex reasoning**.
> - **Synergy across Complementary Modalities:** The combination of Single-Image (SI), Multi-View (MV), and Video data is critical for efficiency. As shown in **Fig. 5**, removing SI and MV data causes the most severe performance drop (**-16.4%** overall). Crucially, removing these static modalities even degrades performance on the video-only VSI-Bench. This proves that **static spatial concepts efficiently transfer to and enhance video spatial reasoning**, allowing the model to learn stable spatial representations from static data that are harder to extract from complex video streams alone.
> - **High-Density Information via Rigorous Filtering:** We prioritize signal density over volume through the rigorous filtering pipeline detailed in **Appendix B.1**. We applied strict constraints, including scene/object diversity checks, noise filtering (removing structural elements like walls/floors), and a minimum visibility threshold (**40%**). These constraints eliminated approximately **90%** of the initially generated samples. By training only on **high-quality, unambiguous samples** where the spatial relationship is visually verifiable, **the model learns far more efficiently than it would from a larger, noisier dataset where the training signal is diluted**.
>
> ### Q3: Have you explored or considered more dynamic training curricula?
>
> We sincerely thank the reviewer for this forward-looking and insightful suggestion. We fully agree that while our current fixed sequential structure is effective, moving towards dynamic or cyclic curricula represents the logical evolution of our framework. We explicitly identified this as a key research direction in our **Limitations and Future Work (Appendix E)**, noting that "developing adaptive training frameworks that dynamically adjust learning sequences... could improve efficiency". We are excited to elaborate on how we envision these advanced strategies:
>
> - **The Potential of Cyclical and Spiral Curricula:** We believe the reviewer’s suggestion of **cyclical reintroduction** (periodically revisiting Stage 1 perception tasks during Stage 3 reasoning training) could offer distinct advantages. As the model adapts to abstract reasoning and complex RL rewards in Stage 3, there is a theoretical risk that its low-level localization sharpness might drift. A cyclical schedule would act as a regularizer, **ensuring that the perceptual foundation remains robust even as higher-order cognitive capabilities are expanded**. This mimics human learning, where fundamental skills are continuously practiced alongside advanced problem-solving.
> - **Adaptive Mixing after Bridging the Gap:** Furthermore, we see great promise in a **"Post-foundation Mixing"** strategy. Our current work establishes that a strict order is necessary to initially bridge the "Perception-Reasoning Gap". However, once this bridge is established, switching to a dynamic, mixed-task schedule could help the model generalize more fluidly across tasks. In future work, we aim to explore **metric-driven dynamic scheduling**, where the training curriculum automatically adjusts the ratio of perception and reasoning data based on real-time validation feedback (e.g., increasing Stage 1 sampling if localization accuracy drops), thereby **automating the progression we currently enforce manually**.
>
> While our current work focused on establishing the necessity of the foundational order, we are excited to explore these dynamic schedules in our future work.

---

> ### Author Response · Authors · 2025-11-20
> **Rebuttal - Part V**
>
> ### Q4: Have you investigated whether the more focused attention in SpatialLadder translates to better performance on multi-object relational tasks?
>
> We thank the reviewer for this insightful question. Inspired by your suggestion, we expanded our visual attention analysis  (updated in **Fig. 7** of our paper) to specifically target multi-object relational tasks to verify if the model can effectively ground multiple entities simultaneously. The results affirm that our training successfully reshapes the model's attentional mechanism.
>
> Qualitatively, as visualized in our updated analysis (e.g., the "Table & Trash Can" example), SpatialLadder consistently generates **distinct, simultaneous attention hotspots** for both entities involved in a spatial query, whereas the base model typically exhibits diffuse or singular attention patterns. Quantitatively, SpatialLadder achieves **73.5%** overall accuracy with **37.7%** attention IoU, significantly outperforming the base model (**32.1%** accuracy, **33.8%** IoU) . This advantage is particularly pronounced in multi-object tasks like relative direction, where precise simultaneous attention—reflected by consistently higher attention IoU and lower entropy—contributes to a massive accuracy improvement.
>
> This evidence confirms that **our training successfully reshapes the model's mechanism to precisely and simultaneously attend to multiple targets**.
>
> ### Reference
>
> [1] MMSI-Bench: A Benchmark for Multi-Image Spatial Intelligence. ICCV 2025
>
> [2] Spatial Mental Modeling from Limited Views.
>
> [3] Detecting hallucinations in large language models using semantic entropy. Nature 2024
>
> [4] Does Reinforcement Learning Really Incentivize Reasoning Capacity in LLMs Beyond the Base Model? NIPS 2025
>
> ------
>
> We sincerely thank Reviewer eN7r once again for the thoughtful and constructive feedback, as well as for the recognition of our work. Your insights have motivated us to conduct additional experiments and refine the manuscript, which we believe has further strengthened the contributions of our study. We remain fully available to address any additional questions or suggestions.

---

> > ### Comment · Reviewer_eN7r · 2025-11-27
> >
> > Dear authors
> >
> > Thanks for your detailed rebuttal. Your clarifications and revisions effectively address my concerns, and I am happy to maintain my score.

---

> > > ### Author Response · Authors · 2025-11-27
> > >
> > > Dear reviewer eN7r,
> > >
> > > Thank you very much for the positive feedback and continued support. We are glad that our responses have addressed your concerns. We deeply appreciate your constructive engagement with our work and welcome any further questions or suggestions you may have.

---

### Official Review · Reviewer_d8EG · 2025-10-31

**Soundness:** 2
**Presentation:** 2
**Contribution:** 2
**Rating:** 4
**Confidence:** 5

**Summary:**

This paper addresses the persistent weakness of current Vision Language Models (VLMs) in spatial reasoning in terms of understanding object positions, distances, and relationships in 2D/3D scenes. The authors argue that this limitation stems from the lack of a hierarchical learning process bridging low-level perception and high-level reasoning. To tackle this, they propose a three-stage progressive training framework that mirrors human cognitive development:

1. Stage 1 : supervised fine-tuning on object localisation tasks.

2. Stage 2 : fine-tuning on a new multimodal dataset covering spatial reasoning tasks across images, multi-view setups, and videos.

3. Stage 3 : reinforcement learning with chain-of-thought supervision (GRPO) to strengthen reasoning quality.

They also introduce SpatialLadder-26k, a curated dataset (26k samples) systematically covering spatial reasoning tasks across modalities. The resulting SpatialLadder-3B model (based on Qwen2.5-VL-3B) achieves state-of-the-art performance on spatial reasoning benchmarks,  including VSI-Bench, SPBench-SI and SPBench-MV, outperforming GPT-4o and Gemini-2.0-Flash by some margins.
The authors claim that this demonstrates the value of progressive spatial training for robust spatial intelligence in VLMs.

**Strengths:**

1. The proposed SpatialLadder-26k dataset is carefully curated and covers a wide range of spatial tasks from object localisation to multi-view and video reasoning, providing a systematic and high-quality resource for studying spatial understanding in VLMs.

2. The three-stage progressive framework is conceptually clean and easy to reproduce. It makes sense, in theory, to have training stages the way the authors designed.

3. The paper reports extensive experiments across multiple benchmarks, including new ones introduced by the authors, with consistent metrics, ablations, and qualitative analyses that make the empirical findings clear and verifiable.

4. The resulting model achieves nice gains over comparable open-source and even proprietary models on several spatial reasoning benchmarks, demonstrating that targeted fine-tuning on well-structured spatial data can significantly improve performance.

**Weaknesses:**

1. Despite being framed as “progressive spatial reasoning,” the method largely reduces to a structured fine-tuning pipeline. Most of the gains appear to come from dataset design and task-aligned supervision rather than from a fundamentally new training principle or architectural innovation.

2. The final stage adds negligible improvement, around one to two percent, and therefore does not convincingly demonstrate that the proposed reinforcement-based reasoning step meaningfully enhances spatial understanding beyond what supervised fine-tuning already achieves.

3. While the authors ablate each stage separately, they never compare progressive versus joint training on all data. Without such a baseline, it remains unclear whether the sequential, staged structure is necessary or if a single fine-tuning step on the combined dataset would yield equivalent results.

4. Both the dataset and the evaluation benchmarks share nearly identical task formats and question templates. This overlap makes it difficult to separate genuine reasoning improvements from specialization to the specific spatial QA style used during training.

**Questions:**

1. Have the authors compared progressive stage-wise training against a single-stage joint fine-tuning using all Stage 1 + Stage 2 data to demonstrate that the “progressive” order is indeed beneficial?

2. Given that Stage 3 yields only marginal gains, can the authors clarify what specific reasoning abilities are improved by the GRPO reinforcement phase, beyond formatting or answer consistency?

3. Since the training and evaluation datasets share highly similar spatial QA formats and scene types, how do the authors ensure that the model’s gains reflect genuine spatial reasoning rather than memorization of task patterns?

4. Qwen2.5-VL is a general-purpose multimodal model. Did the authors evaluate whether SpatialLadder retains its original non-spatial capabilities (e.g., captioning or OCR) after fine-tuning? This is assuming the fact that the authors did full fine tuning of the model and not LoRA, which mostly leads to catastrophic forgetting.

5. Following from my previoous question, are all model parameters fine-tuned during the supervised and reinforcement phases, or are parameter-efficient methods (e.g., LoRA) used? How stable is training when applying full-model updates?

6. Given that Stage 1 improves results only slightly, what evidence supports the claim that perceptual grounding through object localisation meaningfully benefits downstream reasoning?

---

> ### Author Response · Authors · 2025-11-20
> **Rebuttal - Part I**
>
> We sincerely thank Reviewer d8EG for the highly detailed review and constructive assessment. We have carefully addressed the reviewer's questions and concerns as follows.
>
> ### W1: Most of the gains appear to come from dataset design and task-aligned supervision.
>
> We acknowledge that the SpatialLadder-26k dataset is indeed carefully designed. However, both the dataset and the training framework are derived from the specific **"Perception–Reasoning Gap"** diagnosis (**Fig. 1**). Our results demonstrate that the performance gains stem from this **fundamental training principle** **rather than simple data accumulation or direct supervision:**
>
> - **Not Simple Data Accumulation**: The effectiveness of our approach is not driven by the quantity of data. As shown in **Tab. 15**, using our strategy with only 26k samples yields **43.9%** on VSI-Bench under SFT. In contrast, competing datasets like SpaceR-151k and Spatial-MLLM-120k, which contain **5-6** times more data, achieve significantly lower SFT performance (**35.1%** and **40.0**%) on the same backbone. This confirms that our performance gain **does not arise from simply adding more data**.
> - **Not Direct Task Alignment:** Crucially, our gains do not come from simple task-aligned supervision. In the controlled study (**Appendix D.4, Fig. 13**), the hierarchical sequence (Perception $\to$ Spatial) achieves **43.9%**, outperforming both the Mixed setting (40.7%) and the Spatial-Only setting (**42.7%**). This distinction is vital: the 6k object localization samples used in Stage 1 **do not belong to the downstream reasoning task**. If the gains were merely from task alignment, adding these non-reasoning samples shouldn't help, and certainly shouldn't outperform direct training on reasoning tasks. This demonstrates that the model does not benefit from “task-aligned supervision,” but rather from the fundamental principle we follow: a model must first establish perceptual foundations before performing reliable spatial reasoning.
> - **OOD Generalization:** Finally, if the gains were solely due to dataset engineering or overfitting to specific QA templates, the model would fail on unseen domains. However, as detailed in our response to **W4 & Q3**, our model demonstrates robust generalization on strictly out-of-domain benchmarks like **MMSI-Bench (+2.7%)** and **MindCube (+5.8%)**. These benchmarks feature entirely different QA formats and scene types. This confirms that our progressive framework **cultivates generalizable spatial logic rather than merely exploiting dataset-specific biases or supervision shortcuts**.

---

> ### Author Response · Authors · 2025-11-20
> **Rebuttal - Part II**
>
> ### W2 & Q2: does not convincingly demonstrate that the proposed reinforcement-based reasoning step meaningful; can the authors clarify what specific reasoning abilities are improved by the GRPO reinforcement phase?
>
> We thank the reviewer for this insightful question. While the absolute improvement of the final stage may appear modest (+2.1%), our experimental evidence demonstrates that this stage is essential for surpassing the inherent performance ceiling of SFT (Stage 2). Beyond accuracy gains, it fundamentally reshapes the model’s reasoning dynamics, enabling the emergence of more consistent, confident, and higher-quality reasoning chains:
>
> - **Surpassing the SFT Ceiling:** To rigorously determine whether the gains in Stage 3 stem from the RL algorithm itself or simply from extended SFT training steps, we conducted a controlled experiment comparing **Extended SFT** (adding an extra epoch to Stage 2) against our Stage 3 RL. We have updated our paper to include the experiments in **Tab. 3**, highlighted in blue:
>
>   | Model                               | VSI-Bench | SPBench-SI | SPBench-MV | Avg.                         |
>   | ----------------------------------- | --------- | ---------- | ---------- | ---------------------------- |
>   | Qwen2.5-VL-3B (Backbone)            | 29.4      | 40.3       | 36.6       | 38.8                         |
>   | + Stage 1-2 SFT Training            | 43.9      | 68.5       | 69.9       | 60.2                         |
>   | + Stage 1-2 (2 Epochs) SFT Training | 43.4      | 64.3       | 67.0       | 58.2 **($\downarrow$ 2.0%)** |
>   | + Stage 1-3 Training (Ours)         | **45.7**  | **70.2**   | **70.9**   | **62.3 ($\uparrow$ 2.1%)**   |
>
>   The comparison shows that additional SFT actually leads to a consistent drop in evaluation performance (**60.2% → 58.2%**), indicating that extra supervised training can cause overfitting and degrade results. In contrast, Stage 3 RL not only avoids performance degradation but achieves a substantial improvement (**60.2% → 62.3%**), demonstrating that reinforcement learning **plays a crucial role in the overall performance gains**, which is not achievable with merely SFT.
>
> - **Transforming Exploration into Conviction:** We quantify the fundamental shift in "Reasoning Mode" driven by RL using Semantic Entropy [1] (**Fig. 6**), a metric quantifying model uncertainty by measuring the diversity of semantically distinct response clusters. During the SFT phases (Stages 1-2), the entropy rises (**1.24 $\to$ 1.47**), indicating the model is expanding its reasoning space and learning diverse possibilities. However, Stage 3 RL drastically reduces this entropy (**$\to$ 0.66**), incentivizing the model to collapse the solution space into confident, consistent reasoning paths (Consolidation). This quantitative dynamic demonstrates that RL performs a unique function: it does not merely repeat the training data but actively **refines and stabilizes** the reasoning process, **transitioning the model from broad exploration to focused conviction**, which aligns with the phenomena observed in recent studies [2].
>
> - **Structured Reasoning and Self-Verification:** Regarding the specific reasoning abilities improved, the GRPO phase cultivates **structured reasoning and self-correction behaviors** that go beyond simple formatting or answer consistency. As visualized in **Fig. 8**, the model exhibits **systematic decomposition**, breaking complex spatial queries into logical steps rather than generating unstructured descriptions. Crucially, the model develops **self-verification patterns** that are effectively incentivized by the RL reward structure. These behaviors enhance the transparency and auditability spatial reasoning process that SFT alone fails to consistently establish.

---

> ### Author Response · Authors · 2025-11-20
> **Rebuttal - Part III**
>
> ### W3 & Q1: While the authors ablate each stage separately, they never compare progressive versus joint training on all data; Have the authors compared progressive stage-wise training against a single-stage joint fine-tuning using all Stage 1 + Stage 2 data?
>
> We appreciate the reviewer for emphasizing this critical comparison and fully agree that comparing against a joint training baseline is essential. We respectfully clarify that this exact comparison was conducted and reported in **Appendix D.4 (Fig. 13)**, where we performed an ablation study using identical data to strictly compare progressive training against joint (mixed) training. We highlight two critical findings from this experiment:
>
> - **Progressive Training Significantly Outperforms Joint Training:** The results strongly support our claim: our progressive training strategy achieves the highest performance on VSI-Bench (**43.9%**), outperforming the joint training baseline (“Mixed”, **40.7%**). This demonstrates that simply merging different tasks into a single stage or performing joint fine-tuning (**40.7%**) yields substantially worse results compared to our progressive pipeline (**43.9%**). These findings highlight the **critical importance of training tasks in a stage-wise, progressively structured manner**.
>
> - **Evidence of Task Interference:** Crucially, the mixed training performance (**40.7%**) is even lower than the stage 2 only baseline (**42.7%**) . This indicates that naively combining simple object localization tasks (Stage 1) with complex reasoning tasks (Stage 2) leads to **task interference** . Our progressive structure is therefore necessary: it effectively decouples these conflicting objectives, **establishing a stable perceptual foundation first to facilitate the learning of complex spatial reasoning**.
>
> ### W4 & Q3: Both the dataset and the evaluation benchmarks share nearly identical task formats and question templates; how do the authors ensure that the model’s gains reflect genuine spatial reasoning?
>
> We thank the reviewer for raising this critical question regarding potential overfitting to specific QA templates. We acknowledge the similarity in task formats for in-domain benchmarks; however, to explicitly ensure that our model’s gains reflect **genuine spatial reasoning** rather than simple pattern matching, we relied on rigorous OOD evaluations across diverse task paradigms. We present evidence from both existing benchmarks and two new, highly challenging benchmarks (**MMSI-Bench** [3] and **MindCube** [4]) (updated in **Tab. 2** of our paper) to demonstrate that our model generalizes effectively to entirely different reasoning logics.
>
> - **Evidence from Existing OOD Experiments:** First, we highlight the evidence from existing benchmarks presented in **Tab. 2**. Unlike our training data which focuses on specific spatial relations, **CV-Bench** covers traditional vision tasks with completely different QA phrasing and objectives, utilizing scenes from **ADE20K**, **COCO**, and **Omni3D**. Similarly, ViewSpatial-Bench emphasizes viewpoint understanding. Our model achieves consistent gains on CV-Bench (**+3.1%**) and ViewSpatial-Bench (**+8.6%**), providing strong initial evidence that **the learned capabilities are not restricted to specific scene types or question templates**.
>
> - **Generalization on MMSI-Bench and MindCube:** To definitively rule out template overfitting, we further evaluated SpatialLadder on **MMSI-Bench** and **MindCube**, which differ radically from our training data in both domain and reasoning logic:
>
>   - **MMSI-Bench:** This benchmark aggregates data from 8 diverse sources, including **nuScenes, Waymo** (autonomous driving), and **Ego4D** (egocentric robotics), including dynamic situational awareness tasks completely distinct from our static spatial QA templates.
>
>   - **MindCube:** This benchmark tests complex spatial mental rotation and manipulation, requiring abstract spatial logic that cannot be solved by simple pattern-matching of "A is left of B" templates, with scenes from **DL3DV-10K, WildRGB-D, and ArkitScenes**, covering a wide range of **wild indoor/outdoor** RGB-D data.
>
>     As shown in the table below, SpatialLadder-3B consistently outperforms the base model on these distinct domains:
>
>     | Method               | MMSI-Bench | MindCube |
>     | -------------------- | ---------- | -------- |
>     | **Qwen2.5-VL-3B**    | 26.5       | 37.6     |
>     | **SpatialLadder-3B** | **29.2**   | **43.4** |
>     | **Improvement**      | **+2.7**   | **+5.8** |
>
>     The consistent gains across diverse scenes (MMSI, **+2.7%**) and complex mental reasoning tasks (MindCube, **+5.8%**) strongly suggest that our progressive framework teaches the model **abstract spatial logic** **rather than exploiting specific QA shortcuts or domain textures**.

---

> ### Author Response · Authors · 2025-11-20
> **Rebuttal - Part IV**
>
> ### Q4 & Q5: Did the authors evaluate whether SpatialLadder retains its original non-spatial capabilities (e.g., captioning or OCR) after fine-tuning; How stable is training when applying full-model updates?
>
> We thank the reviewer for these critical questions regarding our training methodology and its impact on model robustness. We address the choice of fine-tuning strategy, training stability, and the preservation of non-spatial capabilities in two key aspects.
>
> - **Rationale for Full-Parameter Tuning and Training Stability:** We explicitly confirm that **full-parameter fine-tuning** was employed throughout all training stages. For spatial reasoning tasks that require reshaping internal representations (as shown in **Fig. 7**), we believe full fine-tuning is necessary. Regarding training stability, our full-model updates proved to be highly stable. As shown in **Fig. 4** of the main paper, the Stage-3 RL training exhibits low reward variance and steadily increasing reward curves, demonstrating smooth convergence. To further substantiate this, we have added **training-loss curves for the SFT phases (Stages 1–2) in the Appendix C.5 (Fig. 10)**, which consistently show stable decreasing loss without spikes or divergence, confirming that **our full-parameter approach is mathematically stable**.
>
> - **Preservation of General Capabilities:** To rigorously assess the risk of catastrophic forgetting, we deliberately evaluated the model on **MMBench** [5] and **MMMU** [6], two comprehensive benchmarks that assess broad multimodal competence, including OCR, coarse perception, and multiple subjects knowledge, rather than relying on narrow evaluations like simple captioning. We have updated our paper to include the experiments in **Tab. 4**, highlighted in blue:
>
>   | Method           | MMBench-FP-S (incl. OCR) | MMBench-CP | MMBench-Overall | MMMU (Multiple-Subjects) |
>   | ---------------- | ------------------------ | ---------- | --------------- | ------------------------ |
>   | Qwen2.5-VL-3B    | 88.2                     | 87.9       | 83.3            | 48.3                     |
>   | SpatialLadder-3B | 87.4                     | 87.2       | 82.4            | 47.1                     |
>   | Delta            | -0.8                     | -0.7       | -0.9            | -1.2                     |
>
>   The results demonstrate that, our model maintains high proficiency across these general domains. The performance dip is marginal: specifically, we observe only a **0.9%** decrease on MMBench (Overall) and a **1.2%** decrease on MMMU compared to the base model. Given the substantial gains in spatial reasoning (e.g., **+20.8%** over GPT-4o on VSI-Bench), this negligible trade-off confirms that our progressive training **effectively enhances spatial intelligence while preserving the model’s original captioning, OCR, and broader multimodal foundations**.

---

> ### Author Response · Authors · 2025-11-20
> **Rebuttal - Part V**
>
> ### Q6: what evidence supports the claim that perceptual grounding through object localisation meaningfully benefits downstream reasoning?
>
> We thank the reviewer for this insightful question regarding the specific contribution of Stage 1. While its isolated numerical gain in the ablation study may appear modest (+1.2%), we respectfully argue that Stage 1 plays a foundational role in building robust spatial intelligence, supported by multidimensional evidence spanning cross-task transfer, internal representation dynamics, training stability, and empirical diagnosis.
>
> - **Causal Evidence from Cross-Task Transfer:** We respectfully argue that the nature of the improvement is more significant than its magnitude. The data used in Stage 1 consists solely of object localization tasks and contains **zero spatial-reasoning QA pairs**. If perceptual grounding were not mechanistically linked to reasoning, this stage should contribute nothing (or noise) to downstream performance. The fact that adding purely perceptual supervision yields a consistent performance gain (**+1.2%**) provides strict causal evidence: it proves that **"internalizing localization" is a prerequisite mechanism that directly facilitates complex reasoning**, validating our hierarchical hypothesis.
> - **Sharpening Visual Attention:** Beyond accuracy, Stage 1 meaningfully reshapes the model’s internal visual representations. As shown in **Fig. 7**, the visual attention becomes significantly more focused on task-relevant objects after our training. Quantitatively, the attention IoU improves from **33.8%** to **37.7%**, and the visual attention entropy decreases from **0.193** to **0.176**. This confirms that perceptual grounding enhances the model's foundational ability to **selectively attend to spatially critical regions, reducing noise** during the subsequent reasoning process.
> - **Expanding the Reasoning Space for RL:** Crucially, Stage 1 plays a vital role in expanding the model's exploration capabilities. As analyzed via Semantic Entropy in **Fig. 6**, the entropy increases during Stage 1. This indicates that perceptual grounding enables the model to break away from initial misconceptions and expand its semantic search space, effectively establishing a broader exploration space. This expanded exploration space serves as the essential substrate for the subsequent RL stage, **providing the necessary diversity for the policy to eventually converge onto optimal, high-quality reasoning trajectories**.
>
> - **Stabilizing Optimization Dynamics:** Stage 1 is also crucial for training stability. As illustrated in **Fig. 4 (Middle)**, the model trained with the full progressive pipeline (including Stage 1) exhibits the most significant reduction in reward variance and the smoothest convergence curve compared to ablated variants. Removing the perceptual foundation leads to higher training variance, indicating that Stage 1 **stabilizes the optimization landscape, ensuring that subsequent reasoning training converges efficiently and robustly**.
> - **Validation from Empirical Diagnosis:** Finally, these findings are consistent with our initial diagnostic experiments (**Fig. 1** and **Appendix A**). We observed that simply providing ground-truth object bounding boxes (i.e., simulating the output of Stage 1) boosts the base model’s accuracy from **36.5%** to **41.5%** (**+5.0%**). This diagnostic result aligns with our training outcomes, confirming that the **model’s performance is inherently bottlenecked by inadequate perception**, which Stage 1 effectively resolves.
>
> ### Reference
>
> [1] Detecting hallucinations in large language models using semantic entropy. Nature 2024
>
> [2] Does Reinforcement Learning Really Incentivize Reasoning Capacity in LLMs Beyond the Base Model? NIPS 2025
>
> [3] MMSI-Bench: A Benchmark for Multi-Image Spatial Intelligence.
>
> [4] Spatial Mental Modeling from Limited Views. ICCV 2025
>
> [5] MMBench: Is Your Multi-modal Model an All-around Player? ECCV 2024
>
> [6] MMMU: A Massive Multi-discipline Multimodal Understanding and Reasoning Benchmark for Expert AGI. CVPR 2024
>
> -----
>
> We sincerely thank Reviewer d8EG again for the constructive feedback. These insights have motivated us to conduct additional experiments and provide clarifications that we believe have significantly strengthened our manuscript. We remain fully available to address any further questions.

---

### Official Review · Reviewer_fKYQ · 2025-11-01

**Soundness:** 2
**Presentation:** 3
**Contribution:** 2
**Rating:** 4
**Confidence:** 4

**Summary:**

This paper introduces SpatialLadder, a progressive training framework for enhancing spatial reasoning in vision-language models (VLMs). The approach builds on a new dataset, SpatialLadder-26k, which includes 26k multimodal samples (object localization, single-image, multi-view, and video reasoning). The model is trained in three progressive stages: (1) supervised fine-tuning for perception (object localization), (2) spatial understanding through seven spatial dimensions (direction, distance, size, etc.), and (3) reinforcement learning with GRPO for complex reasoning. The resulting SpatialLadder-3B model achieves strong performance on multiple spatial reasoning benchmarks (VSI-Bench, SPBench-SI, SPBench-MV), outperforming GPT-4o and Gemini-2.0-Flash on average by 20.8% and 10.1%, respectively, with moderate out-of-domain generalization gains (7.2%).

**Strengths:**

- Extensive experimental validation, including ablations, training dynamics, attention visualization, and both in-domain and out-of-domain benchmarks.

- The dataset is clearly defined and covers complementary modalities; the pipeline is systematic and reproducible.
- Empirically strong; achieves state-of-the-art results on spatial reasoning tasks without architectural changes.

**Weaknesses:**

- The novelty is limited. The method primarily combines known strategies (curriculum learning, GRPO-based RL, multimodal fine-tuning) rather than introducing a new algorithmic insight or theoretical formulation.

- The claimed “progressive construction of spatial intelligence” is not quantitatively justified beyond accuracy gains, no analysis on intermediate representation transfer or reasoning trajectory quality.

- The reinforcement learning component is weakly analyzed: reward sensitivity, stability, and ablations on the GRPO configuration are missing.

- Comparisons to alternative architectures (e.g., InternVL, LLaVA-Next) with equivalent compute budgets are absent, weakening generality claims.

- The dataset scale (26k) is relatively small and domain-constrained (indoor ScanNet scenes). This limits claims of “general spatial intelligence.”

**Questions:**

- How much of the performance gain comes from the new data versus the progressive training schedule?

- Is Stage 3 RL critical, or can comparable performance be achieved by longer supervised training?

- How sensitive is performance to the stage ordering or reward scaling?

- Were out-of-domain benchmarks seen or related to the training domains (e.g., ScanNet overlap)?

**Details Of Ethics Concerns:**

No major ethical risks. The dataset relies on public 3D scans and video sources. The paper follows standard academic ethics; minor concern may involve redistribution of third-party datasets.

---

> ### Author Response · Authors · 2025-11-20
> **Rebuttal - Part I**
>
> We sincerely thank Reviewer fKYQ for the highly detailed review and constructive assessment. We have carefully addressed the reviewer's questions and concerns as follows.
>
> ### W1: About novelty. The method primarily combines known strategies.
>
> We respectfully clarify that our contribution is not merely a combination of established techniques (SFT, RL), but rather the **identification of a critical bottleneck in current VLMs** and the design of a **systematic framework** specifically engineered to resolve it. While the optimization tools are standard, the architecture of the learning process is novel and essential.
>
> - **Quantifying the "Perception-Reasoning Gap":**  Unlike existing works that treat spatial reasoning as a monolithic capability, we provide a quantitative diagnosis identifying the "Perception-Reasoning Gap" as a primary bottleneck in current VLM spatial reasoning.  As shown in our preliminary analysis (**Fig. 1**), simply injecting perceptual hints boosts the base model's accuracy from **36.5%** to **46.0%** without any model updates. This is a key insight: models often possess the reasoning logic but **lack the perceptual grounding to activate spatial intelligence**.
> - **Structural Decomposition vs. Naive Combination:** Guided by this diagnosis, SpatialLadder is not a generic application of curriculum learning. Instead, it systematically decouples spatial intelligence into a functional hierarchy: **Perception (Stage 1) $\to$ Understanding (Stage 2) $\to$ Complex Reasoning (Stage 3)**. This differs from standard multimodal fine-tuning. We do not simply feed data from easy to hard; **we explicitly align the training stages with the inherent dependencies of spatial processing**, ensuring the model can "see" (localize) before it tries to "measure" (spatial understanding) or "deduce" (reasoning).
> - **Validation of the Framework:** Our ablation study in **Appendix D.4 (Fig. 13)** validates that this structural design is essential. Under a strict control setting with identical SFT data volume, our progressive framework achieves **43.9%** on VSI-Bench, consistently outperforming a naive mixed-data training strategy (**40.7%**). This confirms that our performance gains stem from the **structured formulation derived from our core insight, rather than merely from data aggregation or naive method combination**.
>
> ### W2: The claimed “progressive construction of spatial intelligence” is not quantitatively justified beyond accuracy gains.
>
> We appreciate the reviewer's emphasis on quantitative validation. Beyond accuracy, we provide in-depth quantitative analyses in **Section 4.4** that explicitly track the evolution of the model's internal states and reasoning behaviors, confirming the effectiveness of our progressive framework.
>
> - **Quantifying Intermediate Representation Transfer:** To verify that perceptual foundations are successfully established and transferred to reasoning tasks, we analyzed the model's visual attention layers (**Fig. 7**). Quantitatively, compared to the base model, SpatialLadder achieves a significant increase in Attention IoU (**33.8% $\to$ 37.7%**) and a decrease in Attention Entropy (**0.193 $\to$ 0.176**). The attention maps shift from diffuse patterns to precise, object-centric focus, proving that **the intermediate representations are now spatially grounded**.
> - **Quantifying Reasoning Trajectory Evolution:** We utilize Semantic Entropy [1] (**Fig. 6**), a metric quantifying model uncertainty by measuring the diversity of semantically distinct response clusters, to quantitatively monitor the "progressive" nature of the learning process across stages. The entropy curve shows a distinct **"Rise-then-Fall"** trajectory. It rises (**1.24 $\to$ 1.47**) during Stages 1-2, indicating the expansion of the solution space (exploration). It then drops sharply to **0.66** in Stage 3 (RL), marking the convergence towards confident reasoning, which aligns with the phenomena observed in recent studies [2].
> - **Reasoning Quality:** Qualitative analysis (**Fig. 8**) further confirms that the model **develops structured metacognitive behaviors**, such as self-verification and error correction, which are absent in the base model.

---

> ### Author Response · Authors · 2025-11-20
> **Rebuttal - Part II**
>
> ### W3 & Q3: The reinforcement learning component is weakly analyzed; How sensitive is performance to the stage ordering or reward scaling?
>
> We sincerely appreciate the reviewer for raising this insightful point. This suggestion prompted us to conduct a more comprehensive analysis of the RL component, which has significantly strengthened the completeness and reproducibility of our work.
>
> - **Sensitivity to Stage Ordering:** As detailed in our response to Weakness 1 and **Appendix D.4 (Fig. 13)**, the performance is indeed **highly sensitive** to the training order. This sensitivity is a positive finding: it empirically validates our central hypothesis that establishing perceptual foundations (Stage 1) is a prerequisite for effective spatial reasoning, thereby **justifying the necessity of our hierarchical framework**.
>
> - **RL Configuration Strategy:** We respectfully clarify that the core contribution of this work is the **SpatialLadder-26k dataset** and the **progressive training framework**, rather than the proposal of a new RL algorithm. To strictly isolate the source of our performance gains, we intentionally aligned our GRPO configurations with established standards (e.g., DeepSeek-R1, VLM-R1, Vision-R1) [3-5]. By keeping the RL hyperparameters consistent with community baselines, we ensure that the observed improvements (e.g., +20.8% over GPT-4o) stem purely from our **progressive framework design** and **data quality**, rather than from hyperparameter engineering or specific RL tuning tricks.
>
> - **Sensitive to RL Configuration:** However, following the reviewer's suggestion, to ensure the **completeness and reproducibility** of our experiments, we conducted additional experiments to analyze the sensitivity of the RL configuration (specifically KL weight and reward scaling). We have updated our paper to include the experiments in **Appendix D.5 (Tab. 16)**, highlighted in blue:
>
>   | $\beta_\text{KL}$ | $w_\text{format}$ | $w_\text{accuracy}$ | VSI-Bench       |
>   | ----------------- | ----------------- | ------------------- | --------------- |
>   | 0.01              | 1.0               | 1.0                 | **45.7** (Ours) |
>   | 0.01              | 0.8               | 0.2                 | 44.9            |
>   | 0.01              | 0.2               | 0.8                 | 45.3            |
>   | 0.04              | 1.0               | 1.0                 | 45.2            |
>
>   As shown in the above below, while our default configuration yields optimal results, **the model exhibits strong robustness to varying hyperparameters**, maintaining consistently high performance across different settings. We empirically observed that the format reward consistently converges rapidly in early training stages. Given this premise, the robustness to reward weights is inherent to the GRPO mechanism: its **group-based advantage normalization effectively cancels out the absolute scaling magnitude** of the accuracy reward once the format reward stabilizes.
>
> - **RL Stability:** Finally, we reaffirm that our progressive stages significantly enhance training stability. As shown in **Fig. 4**, the full SpatialLadder model exhibits the smoothest convergence and lowest reward variance compared to ablated variants, demonstrating that **our framework effectively stabilizes the RL training phase**.
>
> ### W4: Comparisons to alternative architectures with equivalent compute budgets are absent.
>
> We sincerely appreciate the reviewer's constructive suggestion regarding architectural generality. To address this, we expanded our evaluation to include **both** architectures mentioned by the reviewer (**InternVL-2.5** [6] and **LLaVA-Next-Video** [7]) validating our framework beyond the Qwen series. We reproduced the key ablation study (comparing Joint Training vs. Our Sequential Training) on these alternative backbones. We have updated our paper to include the experiments in **Fig. 9**, highlighted in blue:
>
> | Model Architecture      | Base Model | + Joint Training | + Sequential Training | Improvement (Seq. vs. Joint) |
> | ----------------------- | ---------- | ---------------- | --------------------- | ---------------------------- |
> | **InternVL-2.5-2B**     | 23.4       | 38.9             | **41.7**              | **+2.8**                     |
> | **LLaVA-Next-Video-7B** | 34.9       | 42.9             | **44.5**              | **+1.6**                     |
>
> The results demonstrate that our progressive training strategy systematically outperforms the Joint Training baseline on both InternVL (**+2.8%**) and LLaVA-Next (**+1.6%**). This confirms that the "Perception $\rightarrow$ Reasoning" bottleneck is a general phenomenon in VLMs, and our hierarchical solution is a **generalizable methodology** rather than a model-specific optimization.

---

> ### Author Response · Authors · 2025-11-20
> **Rebuttal - Part III**
>
> ### W5 & Q4: The dataset scale is relatively small and domain-constrained; Were out-of-domain benchmarks seen or related to the training domains?
>
> We thank the reviewer for raising this crucial concern. Ensuring that our model acquires generalized spatial intelligence rather than merely memorizing indoor layouts was a central consideration of our work.
>
> To address this, we clarify the strict data separation in our existing results and provide new experimental evidence on two challenging OOD benchmarks (**MMSI-Bench** [8] and **MindCube** [9]) (updated in **Tab. 2** of our paper) to demonstrate robustness across diverse domains.
>
> - **Clarification on Data Overlap and Task Diversity:** We acknowledge that SPAR-Bench and ViewSpatial-Bench incorporate a subset of scenes from ScanNet. However, we enforce a strict scene-level separation: SpatialLadder-26k is constructed exclusively from the ScanNet *training* split, while evaluation is performed on the *validation* split. There is zero data leakage. More importantly, **CV-Bench** consists entirely of scenes from **ADE20K, COCO, and Omni3D**, covering diverse indoor/outdoor environments completely unrelated to ScanNet. The **+3.1%** improvement on CV-Bench already serves as **strong evidence of generalization beyond our training domain**.
>
> - **Generalization on MMSI-Bench and MindCube:** To definitively answer the concern about "general spatial intelligence," we evaluated SpatialLadder on two additional, highly challenging OOD benchmarks that differ significantly from our training data:
>
>   - **MMSI-Bench:** This benchmark aggregates data from 8 diverse sources, including **nuScenes, Waymo** (autonomous driving), and **Ego4D** (egocentric robotics). It forces the model to reason in entirely unseen **outdoor and dynamic environments**, with QA formats completely distinct from our templates.
>   - **MindCube:** This benchmark evaluates complex spatial cognition (e.g., mental rotation) using scenes from **DL3DV-10K, WildRGB-D, and ArkitScenes**, covering a wide range of **wild indoor/outdoor** RGB-D data. It tests deep spatial understanding beyond simple localization.
>
>   As shown in the table below, SpatialLadder-3B consistently outperforms the base model on these distinct domains:
>
>   | Method               | MMSI-Bench | MindCube |
>   | -------------------- | ---------- | -------- |
>   | **Qwen2.5-VL-3B**    | 26.5       | 37.6     |
>   | **SpatialLadder-3B** | **29.2**   | **43.4** |
>   | **Improvement**      | **+2.7**   | **+5.8** |
>
>   The consistent gains across diverse scenes (MMSI, **+2.7%**) and complex mental reasoning tasks (MindCube, **+5.8%**) strongly suggest that our progressive framework teaches the model **abstract spatial logic** that **generalizes effectively beyond the specific textures and layouts of indoor ScanNet scenes**.
>
> ### Q1: How much of the performance gain comes from the new data versus the progressive training schedule?
>
> We thank the reviewer for this critical question. To explicitly quantify how much performance stems from the **new data** versus the **progressive training schedule**, we synthesized the experimental results from **Tab. 15** (Dataset Comparison) and main results into a unified comparison matrix below. The results demonstrates that both components are core contributions of our work and are **complementary rather than independent**.
>
> We compare our SpatialLadder-26k against significantly larger baselines (SpaceR-151k and Spatial-MLLM-120k) on VSI-Bench. All "Naive Joint SFT" baselines use Qwen2.5-VL-3B as the backbone:
>
> | Training Strategy           | SpatialLadder-26k | SpaceR-151k | Spatial-MLLM-120k |
> | --------------------------- | ----------------- | ----------- | ----------------- |
> | Navie Joint SFT (Baseline)  | 40.7              | 35.1        | 40.0              |
> | Three-Stage Training (Ours) | **45.7**          | -           | -                 |
>
> The results demonstrates that:
>
> - **Gain from Data Quality (Horizontal Comparison):** Even when stripped of our progressive strategy (using standard joint training), **SpatialLadder-26k (40.7%)** outperforms SpaceR-151k (35.1%) and Spatial-MLLM-120k (40.0%), demonstrating exceptional **data efficiency**. Despite being **5-6** times smaller than competing datasets, our dataset's **superior annotation quality and task design provide a stronger foundation for spatial reasoning**.
> - **Gain from Progressive Strategy (Vertical Comparison):** Comparing the baseline Joint SFT (40.7%) with our complete Three-Stage Training (45.7%) reveals a substantial **+5.0% absolute improvement**. This gain arises purely from the **progressive framework** (Sequential SFT + RL). By structuring the learning process (Perception $\to$ Understanding $\to$ Reasoning), we unlock the full potential of the data, achieving a state-of-the-art performance that unstructured training cannot reach.

---

> ### Author Response · Authors · 2025-11-20
> **Rebuttal - Part IV**
>
> ### Q2: Is Stage 3 RL critical, or can comparable performance be achieved by longer supervised training?
>
> We thank the reviewer for this interesting question. To rigorously assess whether the improvements observed in Stage 3 arise from the RL stage itself, rather than being achievable through merely extending the SFT training steps, we conducted a controlled experiment comparing **Extended SFT** (adding an extra epoch to Stage 2) against our Stage 3 RL. We have updated our paper to include the experiments in **Tab. 3**, highlighted in blue:
>
> | Model                               | VSI-Bench | SPBench-SI | SPBench-MV | Avg.                        |
> | ----------------------------------- | --------- | ---------- | ---------- | --------------------------- |
> | Qwen2.5-VL-3B (Backbone)            | 29.4      | 40.3       | 36.6       | 38.8                        |
> | + Stage 1-2 SFT Training            | 43.9      | 68.5       | 69.9       | 60.2                        |
> | + Stage 1-2 (2 Epochs) SFT Training | 43.4      | 64.3       | 67.0       | 58.2 **($\downarrow$ 2.0)** |
> | + Stage 1-3 Training (Ours)         | **45.7**  | **70.2**   | **70.9**   | **62.3 ($\uparrow$ 2.1)**   |
>
> The comparison shows that additional SFT actually leads to a consistent drop in evaluation performance (**60.2% → 58.2%**), indicating that extra supervised training can cause overfitting and degrade results. In contrast, Stage 3 RL not only avoids performance degradation but achieves a substantial improvement (**60.2% → 62.3%**), demonstrating that reinforcement learning **plays a crucial role in the overall performance gains**, which is not achievable with merely SFT.
>
> Furthermore, our analyses in **Fig. 6** (semantic-entropy dynamics) and **Fig. 8** (reasoning-trajectory quality), as detailed in our response to **W2**, further demonstrate that RL is essential not only for boosting accuracy but also for shaping the model’s reasoning process. **RL leads to more consistent, confident, and structurally coherent reasoning paths**.
>
> Stage 3 RL is irreplaceable. It transforms the model from "mimicking answers" (SFT) to "learning to reason" (RL), preventing overfitting and significantly enhancing both performance and reasoning quality.
>
> ### Reference
>
> [1] Detecting hallucinations in large language models using semantic entropy. Nature 2024
>
> [2] Does Reinforcement Learning Really Incentivize Reasoning Capacity in LLMs Beyond the Base Model? NIPS 2025
>
> [3] VLM-R1: A Stable and Generalizable R1-style Large Vision-Language Model.
>
> [4] Vision-R1: Incentivizing Reasoning Capability in Multimodal Large Language Models.
>
> [5] DeepSeek-R1: Incentivizing Reasoning Capability in LLMs via Reinforcement Learning.
>
> [6] Expanding Performance Boundaries of Open-Source Multimodal Models with Model, Data, and Test-Time Scaling.
>
> [7] LLaVA-NeXT-Interleave: Tackling Multi-image, Video, and 3D in Large Multimodal Models
>
> [8] MMSI-Bench: A Benchmark for Multi-Image Spatial Intelligence.
>
> [9] Spatial Mental Modeling from Limited Views. ICCV 2025
>
> -----
>
> We sincerely thank Reviewer fKYQ again for the constructive feedback. These insights have motivated us to conduct additional experiments and provide clarifications that we believe have significantly strengthened our manuscript. We remain fully available to address any further questions.

---

### Author Response · Authors · 2025-11-27
**Summary of Responses to Reviewers**

We thank all reviewers for their valuable feedback and are encouraged that they found our motivation clear and reasonable, and our dataset and training framework high-quality and beneficial to the community (**fKYQ, d8EG, eN7r, g2My**). Reviewers highlighted the significant improvements and state-of-the-art performance across extensive evaluations (**fKYQ, d8EG, eN7r, g2My**), and found our ablation studies and analyses sufficient and convincing (**d8EG, eN7r, g2My**). Here, we provide a high-level summary of the changes we've made to address your concerns, and conclude with an overview of our key contributions.

---

**Here is the summary of updates that we've made to address the reviewers' concerns:**

- **We expand training to diverse model architectures and scales.** (**fKYQ, eN7r, g2My**)
  - We extend our training to **InternVL2.5-2B**, **LLaVA-NeXT-Video-7B** (architectural generalization), and **Qwen2.5-VL-7B** (parameter scalability) (**Fig. 9**).
  - We demonstrate that our dataset and training principles are universally applicable across different model structures and sizes, possessing both structural generalizability and parameter scalability.
- **We validate on new, challenging spatial reasoning datasets.** (**fKYQ, d8EG, eN7r**)
  - We add experiments on **MMSI-Bench** and **MindCube** (**Table 2**).
  - We show that SpatialLadder achieves significant gains on both, proving our method builds general spatial intelligence rather than overfitting to specific scenes or QA templates.
- **We verify the critical role of the RL stage.** (**fKYQ, d8EG, eN7r**)
  - We add a comparative experiment showing that additional SFT cannot match the performance of using RL (**Table 3**).
  - We analyze results alongside semantic entropy (**Fig. 6**) and reasoning chain quality (**Fig. 8**) to verify RL's essential role in performance optimization and logic structuring.
- **We evaluate on general multimodal benchmarks.** (**d8EG, g2My**)
  - We add experiments on **MMBench** (general visual capabilities) and **MMMU** (multi-discipline tasks) (**Table 4**).
  - We demonstrate that our model maintains performance on general benchmarks, effectively avoiding catastrophic forgetting.
- **We analyze RL configurations.** (**fKYQ**)
  - We analyze the impact of KL divergence coefficients and reward scaling on RL training (**Table 16**).
  - We demonstrate robustness to these coefficients and provide a theoretical analysis for GRPO's robustness to reward scaling.
- **We provide additional visual and stability analyses.** (**d8EG, eN7r**)
  - We update attention analysis to qualitatively and quantitatively (via IoU and Entropy) demonstrate focused attention on single/multiple relevant objects (**Fig. 7**). (**eN7r**)
  - We add training curves for Stage 1-2 SFT to verify the stability of full-parameter fine-tuning (**Fig. 10**). (**d8EG**)

----

**The contribution of our work is summarized as follows.**

- **Our work reveals a critical challenge in VLMs—the "perception-reasoning gap,"** where simple perceptual hints significantly boost spatial reasoning performance. This finding empirically identifies inadequate spatial perception as a primary bottleneck preventing effective spatial reasoning.
- **We introduce SpatialLadder-26k,** a comprehensive multimodal dataset with 26,610 samples spanning object localization and spatial reasoning across single-image, multi-view, and video modalities, constructed through a standardized pipeline ensuring systematic coverage and high-quality annotations.
- **We design a three-stage progressive training framework** that systematically builds spatial reasoning capabilities by establishing perceptual foundations, developing spatial understanding, and strengthening complex reasoning through reinforcement learning with verifiable rewards.
- **We demonstrate that our approach yields significant performance improvements,** with SpatialLadder achieving state-of-the-art results on multiple benchmarks while maintaining strong generalization to out-of-domain tasks, validating the effectiveness of progressive spatial learning.

Together, our work provides a principled approach to bridging the perception-reasoning gap, offering a solid foundation and valuable resources for advancing spatial intelligence in VLMs.

---

### Meta-Review · Area_Chair_QGZQ · 2026-01-03

**Summary:**

The key reviewer concerns are:

* Novelty and positioning: framed as 'progressive spatial reasoning' but largely a structured fine-tuning pipeline using known tools (SFT, GRPO) [fKYQ,d8EG]

* Evidence for 'progressive construction': need quantitative analysis of intermediate representations and reasoning-trajectory quality, not just accuracy gains [fKYQ,d8EG]

*  RL stage necessity and stability: is RL critical vs. longer SFT? Sensitivity to stage order and KL/reward scaling; clarity on GRPO configuration [fKYQ,d8EG,eN7r]

*  Progressive vs. joint training and generality: require controlled comparison on identical data and demonstrations across architectures / scales [d8EG,g2My,fKYQ,eN7r]

*  Dataset scale and domain bias: 26k, ScanNet-heavy; risk of template / format overlap; need OOD evidence [fKYQ,d8EG]

*  Retention of general VL capabilities and training stability: risk of catastrophic forgetting under full-parameter FT; clarity on LoRA vs. full FT; stability evidence [d8EG,g2My]

As outlined below, the authors provided new experiments and analyses during rebuttal, directly addressing many raised points. Two reviewers initially lean positive and two lean negative. In my assessment, the rebuttal mitigates a meaningful portion of the initially negative reviewers’ concerns, sufficient to shift the balance toward acceptance despite remaining limitations. I propose accept.

**Reviewer Concerns:**

- Novelty/positioning beyond a 'combo of techniques': The authors center the 'perception–reasoning gap' and a learning-process architecture. Under controls, sequential stage 1–2 SFT outperforms mixed/joint training on identical data, and mixing underperforms spatial-only SFT indicating task interference mitigated by the progressive schedule. This supports a modest contribution in training topology (c.f. new algorithms).

- Evidence for progressive construction: authors quantify internal changes and trajectory dynamics. Attention IoU improves (~4%) and semantic entropy rises through SFT then sharply falls in RL. These observations go some way towards addressing queries concerning intermediate-representation evidence; however, the evidence remains modest.

- RL stage necessity and stability: A controlled comparison attempts to show RL adds value that extra SFT does not. While supportive, the absolute gain (~+2.1 percentage points over Stage 1–2 SFT) is modest and should be interpreted as incremental rather than decisive.

- Progressive vs. joint and scale generality: Authors show that sequential-over-joint advantage replicate across backbones and sizes: InternVL2.5-2B ; LLaVA-NeXT-Video-7B ; Qwen2.5-VL-7B. This goes some way to supporting structural generalisability claims.

- Dataset scale, domain bias, and OOD / template concerns: authors add OOD benchmarks with different domains (MMSI-Bench; MindCube). On broader real-world tasks, performance is somewhat stable on MMBench / MMMU and improves substantially on RoboSpatial-Home (+32.5 overall), providing initial evidence that gains are not template-specific. Concerns regarding forgetting can be considered at least partially mitigated.

Remaining limitations

- Absolute novelty lies in curriculum design and diagnosis rather than new algorithms; however, ablations indicate this structure is causally important.
- Dataset remains moderate and indoor-leaning; OOD results help, however broader real-scene / video diversity would further strengthen claims.
- RL analysis, though expanded, focuses on a few hyperparameters; the work could probe policy diversity and reward shaping, and explore dynamic / cyclic curricula proposed by the authors.

**Reviewer Scores:**

* Reviewer eN7r (8, accept): Praised the core hypothesis, methodology, and analyses; after rebuttal explicitly maintained 8.

* Reviewer g2My (6): Asked for progressive vs. joint ablation and broader model coverage; authors provided both plus general-VL evaluations (RealWorldQA, RoboSpatial-Home, MME-RealWorld-Lite, MMBench/MMMU). Reviewer maintained 6 (accept).

* Reviewer fKYQ (4): Raised novelty, progressive-evidence, RL sensitivity, architectural comparisons, and OOD overlap. The rebuttal added intermediate-representation and semantic-entropy analyses, RL vs. extended SFT, KL/reward robustness, multi-architecture results, and OOD (MMSI, MindCube). A modest upward shift was plausible but unconfirmed.

* Reviewer d8EG (4): Questioned distinctiveness of 'progressive' vs. structured SFT, the magnitude/necessity of RL, joint vs. sequential baseline, template overlap, forgetting, and stability. The authors supplied a joint vs. sequential ablation, RL necessity over extended SFT, OOD/template checks, retention results, and training curves. An upgrade was possible but not confirmed.

Decision recommendation

While absolute novelty resides in the curriculum design and diagnostic framing rather than new algorithms, controlled ablations indicate that this structure is causally of value yielding somewhat consistent, transferable gains. The dataset remains moderate in scale and indoor-leaning. OOD results help, but broader real-scene and video diversity would further strengthen the claims. The RL analysis, though expanded, focuses on a limited set of hyperparameters; future work might probe e.g. dynamic curricula as discussed. These caveats notwithstanding, the empirical evidence across ablations, OOD validations, and multi-architecture replications supports acceptance. While the components are standard, the structured training design is empirically validated, and the rebuttal strengthens the case to potentially shift the balance toward acceptance, despite remaining limitations. I propose accept.

---

### Decision · Program_Chairs · 2026-01-26

Accept (Poster)